# Integrative epigenomic and transcriptomic analysis reveals the requirement of JUNB for hematopoietic fate induction

Xia Chen[1,2,7], Peiliang Wang[2,7], Hui Qiu[2,3,7], Yonglin Zhu [2], Xingwu Zhang [2], Yaxuan Zhang[2], Fuyu Duan[4], Shuangyuan Ding[2], Jianying Guo[5], Yue Huang [6] & Jie Na [2✉]

Human pluripotent stem cell differentiation towards hematopoietic progenitor cell can serve as an in vitro model for human embryonic hematopoiesis, but the dynamic change of epigenome and transcriptome remains elusive. Here, we systematically profile the chromatin accessibility, H3K4me3 and H3K27me3 modifications, and the transcriptome of intermediate progenitors during hematopoietic progenitor cell differentiation in vitro. The integrative analyses reveal sequential opening-up of regions for the binding of hematopoietic transcription factors and stepwise epigenetic reprogramming of bivalent genes. Single-cell analysis of cells undergoing the endothelial-to-hematopoietic transition and comparison with in vivo hemogenic endothelial cells reveal important features of in vitro and in vivo hematopoiesis. We find that JUNB is an essential regulator for hemogenic endothelium specialization and endothelial-to-hematopoietic transition. These studies depict an epigenomic roadmap from human pluripotent stem cells to hematopoietic progenitor cells, which may pave the way to generate hematopoietic progenitor cells with improved developmental potentials.

[1] Tsinghua-Peking Center for Life Sciences, Beijing 100084, China. [2] Center for Stem Cell Biology and Regenerative Medicine, School of Medicine, Tsinghua University, Beijing 100084, China. [3] School of Life Sciences, Tsinghua University, Beijing 100084, China. [4] Guangzhou Women and Children's Medical Center, Guangzhou, China. [5] Center for Reproductive Medicine, Department of Obstetrics and Gynaecology, Peking University Third Hospital, Beijing, China. [6] State Key Laboratory of Medical Molecular Biology, Institute of Basic Medical Sciences, Chinese Academy of Medical Sciences & Peking Union Medical College, Beijing 100005, China. [7] These authors contributed equally: Xia Chen, Peiliang Wang, Hui Qiu. ✉email: jie.na@tsinghua.edu.cn

Human pluripotent stem cell (hPSC) differentiating into hematopoietic progenitor cell (HPC) can recapitulate hematopoiesis in the human embryo and provide a platform for mechanistic studies of HPC fate specification. Blood development in mammalian embryogenesis involves three waves of spatiotemporally distinct hematopoiesis[1,2]. The first and second waves arise in the yolk sac and are considered extra-embryonic hematopoiesis[1]. The first wave is transitory and mainly produces primitive erythrocytes, supporting tissue oxygenation for the growing embryo[1,3]. The second wave gives rise to multipotent progenitors, with erythro-myeloid progenitors (EMPs) and lymphoid-primed progenitors (LMPP), independent of hematopoietic stem cells (HSCs)[4,5]. The third wave is intra-embryonic hematopoiesis, where definitive HSCs emerge from the dorsal aorta of the aorta-gonad-mesonephros (AGM) region[6] and are capable of engrafting adult recipients[7]. In all three waves, HSCs are developed from a group of specialized hemogenic endothelial cells (HECs) via endothelial-to-hematopoietic transition (EHT)[8].

In the in vitro differentiation system, HPC development occurs through sequential stages. HPSCs are induced to mesoderm cells with strong vascular differentiation potential[9]. Endothelial progenitor cells are then induced from these mesoderm cells under the stimulus of VEGF and bFGF[10]. Finally, HPCs are generated from intermediate endothelial cells with hemogenic properties via the EHT process[11]. Genetic and functional studies revealed various signaling pathways and transcription factors (TFs) regulating HPC formation. For example, SCL/TAL1 is essential for the transition from vascular mesoderm cells to hematopoietic endothelium[12], MEIS1 controls hematopoietic endothelium generation via regulating TAL1 and FLI1. In addition, RUNX1 and GATA2 are required for the EHT process and hematopoietic fate specification both in vivo and in vitro[13,14]. However, the global transcriptional control mechanisms underlying HPC formation remain largely obscure.

Chromatin remodeling and histone modification are critical to transcriptional regulation. The chromatin accessibility accurately reflects cell types as the open chromatin regions contain regulatory sequences such as promoters and enhancers, which are amenable for cell type-specific TF binding[15]. Cell fate transition from undifferentiated hPSC to HPC requires extensive chromatin remodeling. However, during HPC specification, the chromatin accessibility landscapes of intermediate progenitors, like mesoderm cells and endothelial progenitor cells, remain poorly characterized. In pluripotent stem cells, many developmental genes controlling cell lineage specifications are frequently marked by both H3K4me3 and H3K27me3, and these genes are termed "bivalent" genes, which are poised for rapid activation upon lineage commitment[16–18]. Dissecting the dynamics of bivalent genes in the HPC differentiation process may reveal potential regulators leading to HPC formation in vivo and in vitro.

Here, we comprehensively analyzed the global gene expression, chromatin accessibility, H3K4me3, and H3K27me3 modifications at different stages during the HPC differentiation process. We also used single-cell RNA-seq (scRNA-seq) and single-cell ATAC-seq (scATAC-seq) to uncover the transcriptome and open chromatin features of subpopulations within ECs and HPCs during the EHT window. Integrative analyses depict the temporal changes in chromatin configuration and gene expression during hPSC to HPC differentiation and enable the identification of regulators that orchestrate HPC generation.

## Results

**Epigenetic profiling of HPC differentiation from hPSC.** To investigate how gene expression and chromatin organization are coordinated during human hematopoietic differentiation in vitro,

a previously described stepwise differentiation protocol[19,20] was employed to induce HPC specification. In this study, WiCell H1 hPSCs were used for hematopoietic cells induction (Fig. 1a). Before differentiation, hPSCs were maintained on vitronectin or Matrigel (BD Biosciences) coated plates (Corning) in E8 medium (STEMCELL Technologies). HPSCs were first treated with BMP4 and CHIR99021 for 3 days to induce vascular mesoderm cells (VMEs), marked by $KDR^+$. The cells were then re-plated and cultured with VEGF and bFGF for 2 days to induce $CD31^+$ $CD34^+$ endothelial progenitor cells (EPCs). Afterwards, SB431542 was added to induce $CD43^+$ $CD34^+$ HPCs (Fig. 1a upper panel). This is a simple and efficient approach to generate large quantities of HPCs. Based on this differentiation system, we harvested the intermediate progenitor cells according to the surface antigen and performed RNA-seq, ATAC-seq, and H3K4me3/H3K27me3 ChIP-seq (Fig. 1a lower panel).

Both principal component analysis (PCA) of RNA-seq datasets and hierarchical clustering analysis of ATAC-seq datasets show that the biological replicates are clustered together (Fig. 1b, c). For ATAC-seq and ChIP-seq, at least 9500 peaks were identified in each cell type (Supplementary Fig. 1). Genomic feature annotation graphs show that about 25%, 40%, and 25% of ATAC-seq peaks are distributed to the intergenic, the promoter, and the intron regions, respectively (Fig. 1d left). For histone modifications, we observed that more than 70% of H3K4me3 peaks are located in the promoter regions, and more than 25% of H3K27me3 peaks are preferably deposited in intergenic regions in VME, EPC, and HPC (Fig. 1d middle and right). Furthermore, our data show that the enrichment of H3K4me3 and open chromatin signals at promoters are generally positively correlated with active transcription, while H3K27me3 signals are associated with gene repression (Fig. 1e). This is consistent with the notion that H3K4me3 and open chromatin are implicated in gene activation[21,22], while H3K27me3 marks reflect the transcription repression[21,23]. The expression and chromatin state of lineage-specific genes are also consistent with previous studies (Fig. 1f). For example, the gene expression levels of pluripotency genes POU5F1 (OCT4) and NANOG are the highest in hPSC and are downregulated in VME, EPC, and HPC (Fig. 1f). Accordingly, their promoter regions are marked by higher levels of H3K4me3 and have more relaxed chromatin in hPSC but not in VME, EPC, and HPC, whereas the H3K27me3 peaks show the reverse trends (Fig. 1f). The histone modifications and chromatin accessibilities of the genes involved in VME specification (such as MIXL1[24] and KDR[25]), EPC formation (such as TAL1[26,27] and ETV2[28]), and HPC cell fate decision (such as GATA2[14,29] and LYL1[30]) also display this pattern (Fig. 1f). Collectively, the above results confirm that we generated high-quality ATAC-seq, ChIP-seq, and RNA-seq datasets for hPSC-HPC differentiation.

**Dynamic change of chromatin states during HPC generation.** Open chromatin regions are amenable for TF binding to initiate a variety of transcription regulation[31]. Next, we systematically evaluated the chromatin accessibility dynamics during HPC differentiation. ATAC-seq read counts in peaks of different stages were merged, normalized, and grouped into six clusters via Mfuzz clustering. We then divided these six clusters into three distinct categories (Fig. 2a and Supplementary Fig. 2). Category I contains chromatin regions that are highly accessible in hPSC and VME, then are closed upon differentiation (cluster1, $n = 7065$; cluster2, $n = 7036$). The chromatin regions of category II are more accessible in VME and EPC (cluster3, $n = 7913$; cluster4, $n = 8796$) than in hPSC and HPC. Category III is characterized by higher accessibility in EPC and HPC (cluster5, $n = 4773$; cluster6, $n = 8118$). We then mapped the signals of H3K4me3

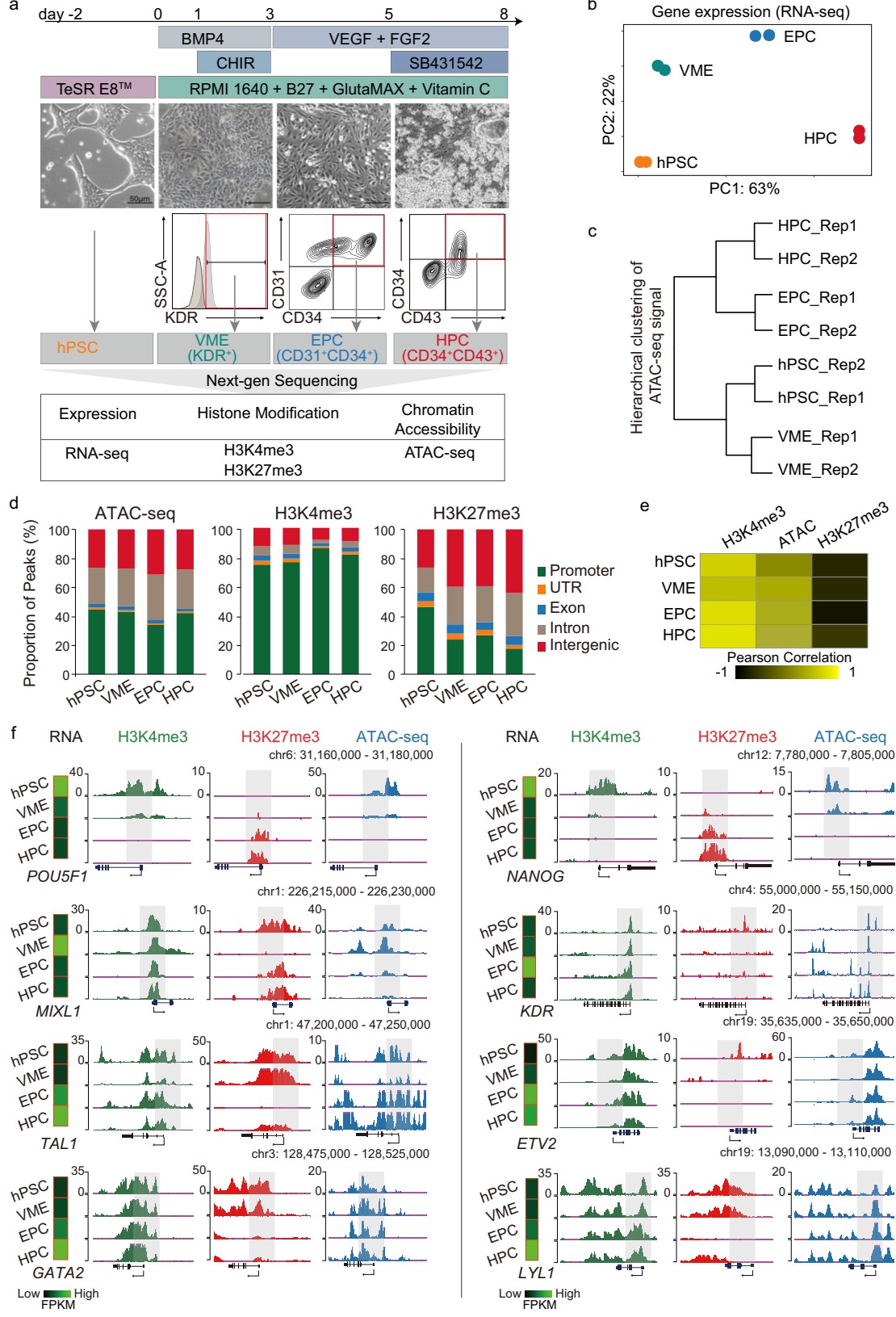

and H3K27me3 along with each open chromatin cluster. The results show that H3K4me3 levels are higher in more accessible regions, whereas H3K27me3 levels are higher in relatively condensed chromatin regions (Fig. 2b). Next, we calculated the expression of genes closest to the ATAC-seq peaks within each cluster, and the results show that the changes in gene expression

follow similar trends with open chromatin dynamics (Fig. 2c). Hence, these results show that the transitory accessible chromatin domains function as cis regulators, such as promoters or enhancers, to facilitate lineage-specific gene expression.

Gene Ontology (GO) analysis is consistent with this notion. Category I genes show enrichment of biological processes

**Fig. 1 Dynamic change of chromatin status during HPC differentiation. a** Schematics showing hematopoietic differentiation process from hPSC (upper panel) and the sample collection strategy for transcriptomic and chromatin state analyses (lower panel). **b** PCA for all differentiation stage RNA-seq replicates. **c** Clustering dendrogram of hPSC, VME, EPC, and HPC replicates for ATAC-seq signals at promoters (TSS ± 2.5 kb). **d** Bar plots showing peak distribution of all differentiation stage ATAC-seq, H3K4me3, and H3K27me3 ChIP-seq data. **e** Heatmap showing the Pearson correlation between gene expression, H3K4me3, H3K27me3 binding, and chromatin accessibility at promoters (TSS ± 2.5 kb) in hPSC, VME, EPC, and HPC. **f** UCSC browser snapshots showing the ChIP-seq and ATAC-seq profiles of representative marker genes at different stages of HPC differentiation. Heatmaps on the left of each panel show the normalized gene expression levels. The view scale of the genome browser is adjusted according to the global data range. hPSC-, VME-, EPC- and HPC- putative promoters are shaded. hPSC human pluripotent stem cell, VME vascular mesoderm cell, EPC endothelial progenitor cell, HPC hematopoietic progenitor cell, Source data are provided as a Source Data file.

involved in pluripotency (such as stem cell proliferation ($p < 1 \times 10^{-5}$)) and early mesoderm development (such as mesoderm development ($p < 1 \times 10^{-5}$)) (Fig. 2d and Supplementary Data), suggesting these genes are activated in hPSC and subsequently silenced to exit pluripotency and facilitate EPC and HPC differentiation. Category II genes are strongly enriched for signaling pathways regulating mesoderm (such as the WNT signaling pathway ($p < 1 \times 10^{-16}$)) and endothelium development (such as the NOTCH signaling pathway ($p < 1 \times 10^{-8}$)) (Fig. 2e and Supplementary Data). Category III genes are related to endothelial (such as endothelium development ($p < 1 \times 10^{-5}$)) and blood lineage cell fate and functions (such as regulation of platelet activation ($p < 1 \times 10^{-6}$)) (Fig. 2f and Supplementary Data), which are vital for EPC and HPC formation.

Next, we used HOMER[32] to search TFs enriched in each open chromatin cluster (Fig. 2g). In category I, accessible regions (clusters 1 and 2) are enriched for OCT4 (POU5F1) and NANOG binding motifs, consistent with their reported roles in pluripotency maintenance. In category II and category III, many endothelial and hematopoietic TFs are enriched. For example, open regions in cluster 4 (EPC high) show motif enrichment for SOX17, which promotes arterial endothelial cell (AEC) specification and is involved in EHT[33]. The binding motif of RUNX1, a master regulator of EHT and hematopoiesis[34], is primarily found in category III open regions, which are highly accessible in HPC. Interestingly, other notable TFs, including the ETS family (ETV1,2,4, and FLI1), which are essential in EC development, and the GATA family, which plays critical roles in HSPC generation, are also enriched significantly during HPC specification (cluster 4, 5, and 6) (Fig. 2g). The above analyses suggest that TFs involved in the processes of HPC specification can be inferred from open chromatin landscape data. Moreover, we can use these ATAC-seq data to identify potential drivers of HPC formation.

**Regulation of bivalent genes during HPC differentiation.** To analyze the chromatin state throughout HPC differentiation, we performed ChromHMM analysis[35] based on the profiles of two histone modifications and identified four chromatin states, which are H3K4me3-only, bivalent, unmarked, and H3K27me3-only states, respectively (Fig. 3a). The annotation results show that the H3K4me3-only regions and bivalent regions are enriched in the transcription start sites (TSS) and promoter regions across all stages (Supplementary Fig. 3a). Although both bivalent and H3K4me3-only regions have similar chromatin accessibilities, the gene expression levels in bivalent regions are significantly lower, consistent with the fact that genes marked by bivalent histone modifications are poised to be activated[36] (Fig. 3b). In contrast, both unmarked and H3K27me3-only regions have low gene expression levels and chromatin accessibility (Fig. 3b).

Bivalent domains are preferentially enriched at promoters of developmental genes[17]. Information about the dynamic change of bivalent genes during differentiation facilitates the identification of many lineage regulators[18,37]. Thus, we analyzed the bivalent

domain profile between successive cell types during the hematopoietic specification process. We found that the number of bivalent domains is highest in hPSC, reflecting more developmental regulators poised in the pluripotent state than other progenitor cell types (Supplementary Fig. 3b).

We next examined the dynamics of bivalent domains and their associated gene expression levels accompanying HPC differentiation. We enumerated the 4 modes of change in bivalent domains and calculated the ratio of each mode (Fig. 3c and Supplementary Fig. 3c). During hPSC differentiation into VME, about 20% of bivalent domains lose H3K27me3 marks but gradually gain H3K4me3 levels and turn into H3K4me3-only regions. Correspondingly, their related genes get activated (Fig. 3d). GO analysis of these activated genes show that the most significant terms include mesoderm development ($p < 1 \times 10^{-7}$) and response to BMP ($p < 1 \times 10^{-8}$) (Fig. 3e and Supplementary Data), which reflect that under the stimulation of BMP4 and CHIR, bivalent genes involved in mesoderm specification (e.g., GATA6, HAND1, and BMP2) are rapidly activated (Fig. 3f). Similarly, from VME to EPC, nearly 20% of bivalent domains turn into H3K4me3-only regions. Prominent GO terms for these genes are related to endothelium development (Fig. 3g, h, and Supplementary Data). Interestingly, genes involved in hematopoietic progenitor cell differentiation are also induced and activated in the EPC stage, suggesting that the hematopoietic program is already primed in the EPCs, as indicated by the loss of H3K27me3 peaks on MYB promoters (Fig. 3i). As a result, only a few bivalent domains (about 7%) turn into H3K4me3-only states from day 5 EPC to day 8 HPC (Supplementary Fig. 3c). These analyses suggest that the key bivalent genes encoding stage-specific regulators are resolved and activated appropriately to promote cell fate transitions during HPC differentiation.

Between every two successive steps, many bivalent domains remain covered by H3K4me3 and H3K27me3 signals, and the related genes remain silenced (Supplementary Fig. 3d–h). These genes are mainly involved in non-hematopoietic lineage commitment, such as embryonic organ development and neuron fate commitment (Supplementary Fig. 3g and Supplementary Data). Thus, the stable bivalent modifications can safeguard the hematopoietic lineage commitment in the in vitro differentiation system. However, HOXA5, HOXA9, and HOXA10, which play critical roles in definitive HSC generation and proliferation[38,39], are also stable bivalent genes throughout the in vitro HPC differentiation (Fig. 3j). In addition, the chromatin around these genes remains largely inaccessible (Fig. 3j), which creates a barrier to their activation.

Collectively, the analyses of bivalent genes provide insights into the unique features of transcriptional and epigenetic regulation of in vitro HPC formation.

**Single-cell analysis reveals subpopulations of ECs and early HPCs.** The EHT is an important event during HPC specification, by which a subset of endothelial cells, termed hemogenic endothelium, lose their spreading EC morphology and round up to

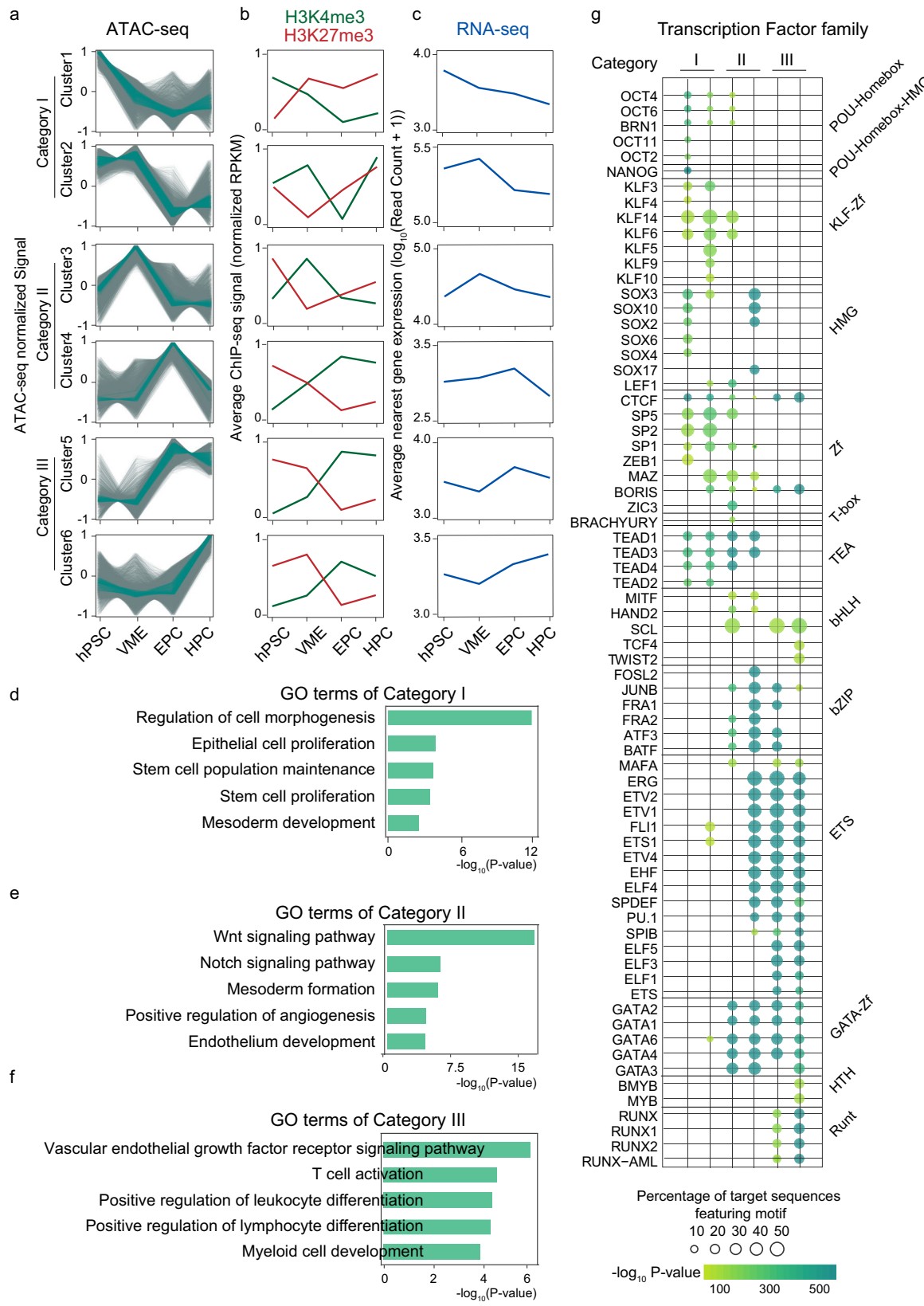

form floating HSPCs. To evaluate the cell diversity and cell-cell interaction during the EHT process in vitro, we performed scRNA-seq and scATAC-seq on differentiation day6 (Fig. 4a). Four distinct cell clusters are distinguished based on their transcriptome (Fig. 4b and Supplementary Fig. 4a, b). The entire population consists of 6.3% epithelial cell (Epi) characterized by the high expression of *CDH1* and *EPCAM*, 31.2% mesoderm cell (Mes) with *HAND1* and *COL3A1* expression, 36.4% endothelial cell (EC) expressing *KDR* and *CD34*, and 26.1% hematopoietic progenitor cell (HPC) with an exclusive expression of *RUNX1* and *GFI1B* (Supplementary Fig. 4a). GO analysis shows that EC and HPC clusters are enriched with endothelial and blood lineage

**Fig. 2 Temporal epigenetic signatures and gene expression pattern during HPC differentiation. a** Mfuzz clustering analysis of ATAC-seq peaks during HPC differentiation. ATAC-seq peaks are clustered into six clusters and grouped into three categories. The line plots depict the standardized ATAC-seq signals, with each gray line representing the signal of the same loci in different cell populations and the green lines representing the values between the upper and lower quartiles of each group. **b** Signals of H3K4me3 and H3K27me3 modifications in each ATAC-seq cluster. **c** Average log-transformed gene expression (read count) for genes closest to the ATAC-seq peaks within each cluster. **d–f** Bar charts showing the statistically over-represented biological processes enriched by GO analysis of different categories of ATAC-seq clusters. p values, Fisher's exact test. **g** The TF motifs enrichment in each ATAC-seq category. The dot size reflects the proportion of sequences containing the specific TF motif, and the color shows the enrichment from Fisher's exact p value. hPSC human pluripotent stem cell, VME vascular mesoderm cell. EPC endothelial progenitor cell, HPC hematopoietic progenitor cell. Source data are provided as a Source Data file.

genes, respectively (Supplementary Fig. 4b and Supplementary Data). Interestingly, oxidative phosphorylation and mitochondria ATP synthesis genes are significantly upregulated in HPCs, indicating that the newly formed HPCs have a highly active energy metabolism (Supplementary Fig. 4b and Supplementary Data). On the other hand, the Mes cluster cells highly express genes related to mesenchyme, extracellular matrix, cartilage, and connective tissue. In contrast, Epi-cluster-specific genes are enriched for the urogenital system and kidney development processes, indicating they possess the gonad-mesonephros tissue features reminiscence of the embryonic hematopoiesis niche (Supplementary Fig. 4b and Supplementary Data). Next, we used Seurat to predict scATAC-seq cell clusters based on the activities of open chromatin-associated genes and found that the scATAC-seq and scRNA-seq clusters are well-matched (Fig. 4c). Moreover, the pseudo-bulk open chromatin profile of the *CD34* based on the scATAC-seq dataset reveals that it is specifically accessible in EC and HPC clusters (Fig. 4d), which confirms the good correlation between scATAC-seq and scRNA-seq data.

HEC expresses both endothelial and hematopoietic genes and is a relatively rare population in vivo[8,40] and in vitro[41]. To identify the HEC population generated in vitro, the annotated EC and HPC populations were extracted, re-normalized, and re-grouped into 5 sub-clusters. Two sub-clusters expressing arterial endothelial genes such as *SOX17*[33], *EFNB2*[42], and *GJA4*[43] were annotated as AEC clusters (Fig. 4e, f, and Supplementary Fig. 4c). Two sub-clusters expressing hematopoietic genes, such as *CLEC11A*[44] were annotated as pre-HPC clusters (Fig. 4f). The sub-cluster expressing endothelial and hematopoietic genes was annotated as the hemogenic endothelium (HE) cluster (Fig. 4f and Supplementary Fig. 4c). Based on the expression level of cell cycle genes, the AEC cluster was further divided into AEC-type I (AEC TI) and AEC-type II (AEC TII) sub-clusters. Similarly, pre-HPC clusters were separated into pre-HPC type I (pre-HPC TI) and pre-HPC type II (pre-HPC TII) sub-clusters (Fig. 4f and Supplementary Fig. 4c). Cell cycle analysis shows that cells in AEC TII, pre-HPC TII, and HE clusters are mostly cycling cells in either the S or G2/M phase (Fig. 4g). Accordingly, cell division-related genes like *BIRC5*, *TPX2*, *CENPE*, and *CDK1* show higher expression in ACE TII, pre-HPC TII, and HE clusters (Fig. 4f). Pseudo-time trajectories show that the HE cluster lies on the interface of AEC and the newly emerged pre-HPC cluster (Fig. 4h), which suggests that hematopoietic cells are derived from AEC via the HE population in vitro. Consistent with this presumption, along with the HPC differentiation trajectory, genes related to AEC development, represented by *GJA4* and *SOX17*, show a gradual decrease, while genes involved in EHT, such as *RUNX1* and *GATA2*, are rapidly increased (Fig. 4i).

Previous studies showed that the interaction between HECs and their neighboring cells plays a pivotal role in HPC generation[8,45,46]. Thus, we performed ligand-receptor analysis to explore the interaction among these cell clusters (Supplementary Fig. 4d). More than 30 ligand-receptor pairs were detected

between HE and other clusters. Among these, TGF-β signaling and NOTCH signaling pathways are significantly enriched (Supplementary Fig. 4d). Notably, NOTCH-JAGs are present in AEC TI and Epi cells, accordant with the previous report that NOTCH is required for the HEC specification from hPSCs[45].

**Comparative analysis of HEC generated in vitro and in vivo.** A recent study showed two temporally and molecularly distinct HEC populations in developing human embryos, at CS10 and CS13, respectively[8]. Furthermore, CS10 HECs are thought to be devoid of definitive HSC potential, while CS13 HECs are thought to be HSC-primed HEC[8]. We compared our scRNA-seq results with recently published CS10 and CS13 HEC scRNA-seq data[8]. The results show that CS10 and CS13 HECs cluster together (Fig. 5a), and the in vitro HEC is more similar to CS10 HEC (Pearson correlation $R = 0.9$) than CS13 HEC (Pearson correlation $R = 0.8$) at the transcriptome level (Fig. 5a, b). We performed k-means clustering analysis of differentially expressed genes between in vitro HEC, CS10, and CS13 HECs and obtained 6 clusters (Fig. 5c). Genes in cluster 1 are upregulated in both CS10 HECs and CS13 HECs, and these genes are related to hypoxia, which is attributed to the hypoxic tissue environment in vivo (Fig. 5c). Cluster 2 genes are upregulated specifically in CS13 HECs, and these genes are mainly involved in endothelium development and artery development. While in vitro HECs and CS10 HECs show less arterial endothelial features but upregulated genes related to cell cycle transition (cluster 4) and DNA replication (cluster 5), indicating that they are in an active proliferation state compared to CS13 HECs (Fig. 5c). The above analyses suggest that culture in low oxygen conditions, promoting arterial endothelial features and cell cycle adjustment, may enhance the potential of in vitro differentiated HECs to form definitive hematopoietic stem cells (HSCs).

**JUNB is a potential regulator of HPC specification.** We then sought to identify previously unknown regulators for HPC differentiation. Motif analysis of differential accessible chromatin sites reveals that TFs responsible for EC and HPC development are mainly enriched in category II and III peaks. To retrieve potential HPC regulators more precisely, we overlapped the TFs whose motifs are enriched in category II and category III peaks with activated bivalent genes (Supplementary Fig. 5), resulting in 16 candidates including several known master regulators of EHT such as RUNX1 *and* GATA2 (Fig. 6a). The expression of bivalent gene *JUNB* is suppressed in hPSC but gradually activated along with HPC specification (Fig. 6a, b), which implies that JUNB may play a role during HPC differentiation. ScRNA-seq data show that *JUNB* is exclusively expressed in EC and HPC populations (Fig. 6c), and its motif is significantly enriched in hemogenic endothelium (HE) populations (Fig. 6d), which further supports our hypothesis that JUNB may be involved in HPC specification. In addition, *JUNB* is also expressed in in vivo HECs (Fig. 6e). Footprint analysis shows that JUNB has a deeper footprint flanking its motifs in EPCs and HPCs than in hPSCs and VMEs,

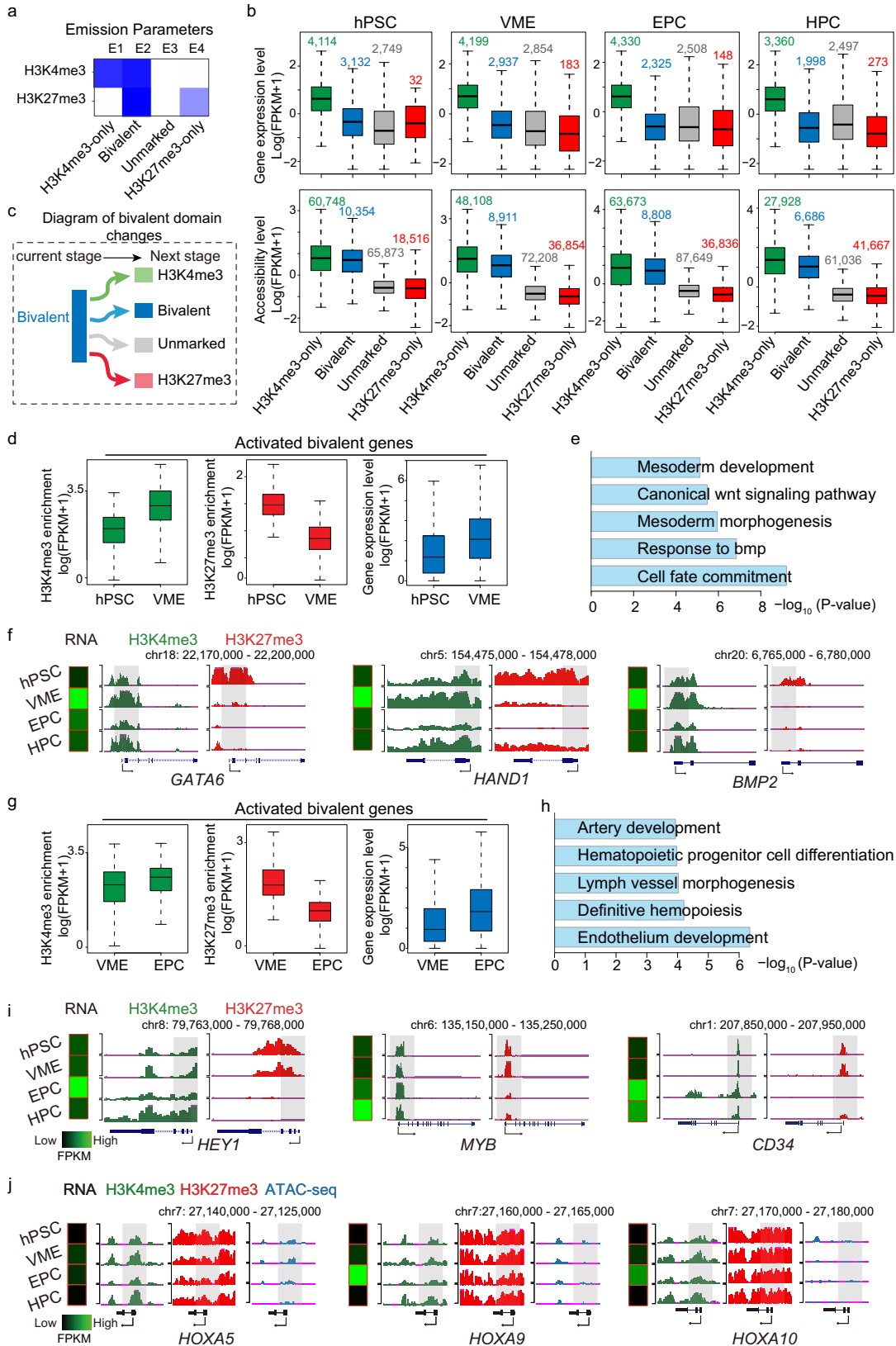

which is similar to the footprint analysis results of three well-known hematopoietic regulators, MYB, MEIS1, and RUNX1. This result indicates that JUNB can directly and preferentially bind to the genome at EPC and HPC stages (Fig. 6f). The above results suggest that JUNB may be a key regulator for EPC to HPC differentiation.

**JUNB deficiency severely impairs HEC and HPC formation.** To find out the role of JUNB in HPC differentiation, we disrupted *JUNB* in HUES8 hESC using the iCRISPR system[47] (Fig. 7a). We selected two mutant lines (KO-1 and KO-2) for further studies. Sanger sequencing and western blot confirmed the deletion of nucleotides at the sgRNA targeting site of *JUNB* and the loss of

**Fig. 3 Dynamic change of histone modification landscape during HPC differentiation. a** Heatmap showing chromatin states classified from ChIP-seq datasets based on ChromHMM algorithm. Each row corresponds to a different histone mark, and each column represents a different chromatin state. Regions with low H3K4me3 or H3K27me3 modifications are labeled as unmarked; regions with both H3K4me3 and H3K27me3 are labeled as bivalent. Darker colors indicate higher probabilities. **b** Boxplots showing corresponding gene expression levels (upper panel) and chromatin accessibilities (lower panel) to each chromatin state at each differentiation stage. Gene/peak number is labeled on the top of each box. **c** Schematic diagram showing the dynamics of bivalent domains during cell state transition. **d** Boxplots showing the dynamics of H3K4me3 (left), H3K27me3 (middle), and gene expression (right) of activated bivalent genes ($n = 419$) during hPSC to VME transition. **e** GO analysis of activated bivalent genes during the hPSC to VME transition. **f** The UCSC browser views showing H3K4me3 and H3K27me3 modification profiles in representative gene loci during hPSC to VME transition. The putative promoters are shaded. **g** Boxplots showing the dynamics of H3K4me3 (left), H3K27me3 (middle), and gene expression (right) of activated bivalent genes ($n = 253$) during VME to EPC transition. **h** GO analysis of activated bivalent genes during VME to EPC transition. **i** The UCSC browser snapshots showing H3K4me3 and H3K27me3 profiles in representative gene loci during VME to EPC transition. The putative promoters are shaded. **j** UCSC genome browser snapshots showing H3K4me3 and H3K27me3 modification profiles and chromatin accessibilities in *HOXA5*, *HOXA9*, and *HOXA10* gene loci. The putative promoters are shaded. Heatmaps in **f**, **i**, **j** showing the expression levels of representative genes. Box boundaries in **b**, **d**, **g** are the 25th and 75th percentiles, the horizontal line across the box is the median, and the whiskers indicate the minimum and maximum values. *p* values, Fisher's exact test. The view scale of the genome browser is adjusted according to the global data range. hPSC human pluripotent stem cell, VME vascular mesoderm cell, EPC endothelial progenitor cell, HPC hematopoietic progenitor cell. Source data are provided as a Source Data file.

JUNB protein expression, respectively (Supplementary Fig. 6a, b). *JUNB* knockout (KO) cells also have normal diploid karyotype (Supplementary Fig. 6c). The pluripotency of *JUNB* KO hPSC is retained as the expression of *OCT4, NANOG*, and *SOX2* are comparable in KO cells and wild-type (WT) cells (Supplementary Fig. 6d). We then differentiated *JUNB* KO and WT hPSCs into HPCs. *JUNB* KO severely inhibits the generation of CD34+CD43+ HPCs on day 8 (Fig. 7b; see Supplementary Fig. 6e for FACS gating strategy). Interestingly, *JUNB* ablation does not reduce the percentage of KDR+ VME cells, CD34+ EPCs, or CD31+/CD144+ ECs (Supplementary Fig. 6f–i; see Supplementary Fig. 6f–i for FACS gating strategy). However, GO analysis shows that *JUNB* KO CD34+ EPCs upregulate mesoderm genes but downregulate endothelial genes compared to WT CD34+ ECs (Fig. 7c and Supplementary Fig. 7a), indicating a skewed transcription signature of CD34+ EPC in *JUNB* KO group. Since CD34+ EPC is a heterogeneous population, and only CD34+ HECs can generate hematopoietic cells[48], we postulated that JUNB might function in in vitro HEC specification. As HEC can be distinguished from endothelial cells based on CD73 and CD184 expression[48], to verify our hypothesis, we counted the percentage of HECs at differentiation day 6 in WT and *JUNB* KO groups, respectively. The results show that the percentage of HEC (CD34+CD73−CD184−) in *JUNB* KO group reduces significantly compared to that in WT group (Fig. 7d; see Supplementary Fig. 7b for FACS gating strategy), which suggests that JUNB plays a role in HEC formation. We analyzed the transcriptomic data of HECs from both *JUNB* KO and WT groups and found that genes (such as *RUNX1, CD44*) functioning in the EHT process are downregulated in *JUNB* KO HECs (Fig. 7e).

To determine how JUNB regulates hematopoietic genes, we studied its genome-wide binding in WT HECs by CUT&Tag[49]. The results reveal that JUNB can bind to the promoters of known key hematopoietic regulators (such as RUNX1, CD44) (Fig. 7f). Next, we checked the change in chromatin accessibility after *JUNB* KO in HECs. Compared to WT HECs, a large portion of ATAC-peaks (20705/60761, 34.1%) are attenuated for two folds (termed as JUNB dependent open regions), and a small portion of ATAC-peaks (8059/60761, 13.2%) gains (termed as JUNB independent open regions) in *JUNB* KO HECs (Supplementary Fig. 7c, d). Motif analysis shows that JUNB and many hematopoietic TF motifs such as ERG, GATA2, and RUNX1 are significantly enriched in JUNB-dependent open regions compared to JUNB independent open regions (Supplementary Fig. 7e). These results suggest that JUNB promotes HEC formation by creating a more accessible chromatin environment for hematopoietic TFs to bind. Based on these results, we

speculated that JUNB might regulate hematopoietic TFs by two mechanisms. Firstly, it directly binds to the prompter of hematopoietic master TFs. Secondly, it opens nearby chromatin to promote the deposition of hematopoietic master TFs.

To determine the roles of JUNB at different stages of hematopoietic specification, we generated inducible JUNB rescue cells (Fig. 7g and Supplementary Fig. 7f). *JUNB* starts to be activated on day 3 during differentiation when the cells are primed to become endothelial progenitor cells and HECs. So, we added DOX from day 3 to day 5 to examine the function of JUNB during HEC formation (Fig. 7h). To examine the role of JUNB in EHT, we added DOX from late day 5 to day 6 (24 h), when EHT is underway (Fig. 7g, h). Re-introducing JUNB during day 3–5 increased the percentage of HECs (about 25%) and HPCs (about 10%) significantly compared to the *JUNB* KO cells (about 15% and 4%, respectively), which confirms the importance of JUNB in HEC and subsequent HPSC formation (Fig. 7i, j). Furthermore, adding JUNB at the EHT window (day 6) also significantly elevates the HPC percentage (Fig. 7j), indicating that HECs can undergo EHT upon JUNB compensation. Taken together, the rescue experiments verify that JUNB is essential for HPC formation by promoting both HEC specification and EHT.

## Discussion
In this study, we systematically profiled the genome-wide chromatin accessibility, H3K4me3, and H3K27me3 modifications and gene expression during sequential stages of hematopoietic fate specification, as well as the single-cell transcriptome and accessible chromatin during the early EHT window. Integrative analysis of the above datasets revealed sequential opening-up of chromatin sites for key TFs regulating hematopoiesis and stepwise epigenetic reprogramming of bivalent genes. The comparative analysis of scRNA-seq data between in vitro and in vivo HEC populations pointed out possible ways to optimize the in vitro HEC protocol. Through the updated understanding of hematopoiesis by these multi-omics data, we identified JUNB as a critical TF for both HEC specialization and the EHT process, which are pivotal to HPC formation.

Many critical developmental genes are both marked by H3K4me3 and H3K27me3 (bivalent state) in pluripotent stem cells[17,50]. Here, we extended the analysis of bivalent genes to VME, EPC, and HPCs. In VME, EPC, and HPCs, bivalent marks only occupy a small portion of the genome compared to H3K4me3 only, H3K27me3 only, and unmarked regions (Supplementary Fig. 3a). Many bivalent genes are inherited from hPSCs. Notably, the bivalent genes activated in the following

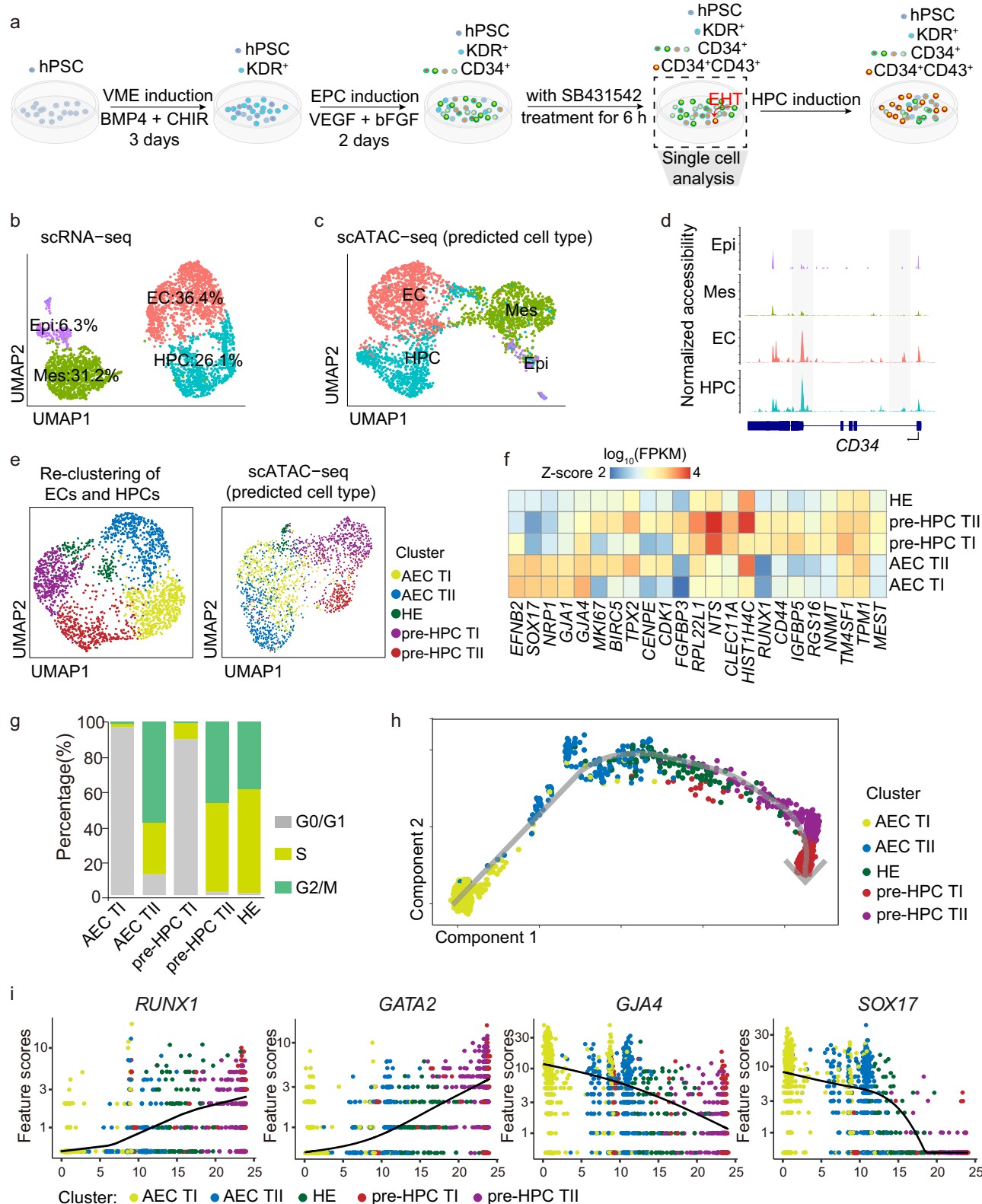

stage accurately reflected lineage choice. From hPSC to VME, the top GOs of activated bivalent genes include mesoderm development and morphogenesis. While from VME to EPC, artery development and hematopoiesis are among the top GOs. From EPC to HPC, only 7% of bivalent genes are activated. This may reflect insufficient activation of HSC master genes in the in vitro differentiation system. Notably, *HOXA5, HOXA9*, and *HOXA10*

are bivalent genes and key TFs for the engraftment potential of definitive HSCs. Their genomic regions remain heavily marked by H3K27me3 signals and are lowly or only transiently expressed during differentiation (Fig. 3j). The bivalent state of *HOXA* genes in EPCs and HPCs may explain why many in vitro generated HPCs have poor engraftment potentials and cannot form lymphoid cell types. This notion is also supported by another

**Fig. 4 ScRNA-seq and scATAC-seq analysis revealed subpopulations of ECs and early HPCs. a** Schematic showing the sample collection for single-cell RNA-seq and ATAC-seq. Dots indicate the different types of cells in the culture dish. **b** UMAP visualization of HPC, EC, Mes, and Epi clusters resulted from scRNA-seq of differentiation day 6 cells. **c** UMAP visualization of HPC, EC, Mes, and Epi clusters resulted from scATAC-seq data. **d** ScATAC-seq peaks around the *CD34* locus. The new emerged open regions are shaded. **e** UMAP plot of scRNA-seq (left) and scATAC-seq (right) of the AEC TI, AEC TII, pre-HPC TI, pre-HPC TII, and HE clusters, derived from re-clustering EC and HPC populations described in **b**. **f** Heatmap showing the expression of top DEGs in AEC TI, AEC TII, pre-HPC TI, pre-HPC TII, and HE populations. **g**, **h** Cell cycle and trajectory analyses of AEC TI, AEC TII, pre-HPC TI, pre-HPC TII, and HE populations. **i** Expression of representative HPC and AEC related genes during the EHT process in AEC TI, AEC TII, pre-HPC TI, pre-HPC TII, and HE populations. hPSC human pluripotent stem cell. VME vascular mesoderm cell, EPC endothelial progenitor cell, HPC hematopoietic progenitor cell, Epi epithelial cell, Mes mesoderm cell, EC endothelial cell, HE hemogenic endothelium, AEC arterial endothelial cell, AEC TI, AEC-type I, AEC TII, AEC-type II, pre-HPC TI pre-HPC type I, pre-HPC TII pre-HPC type II, Source data are provided as a Source Data file.

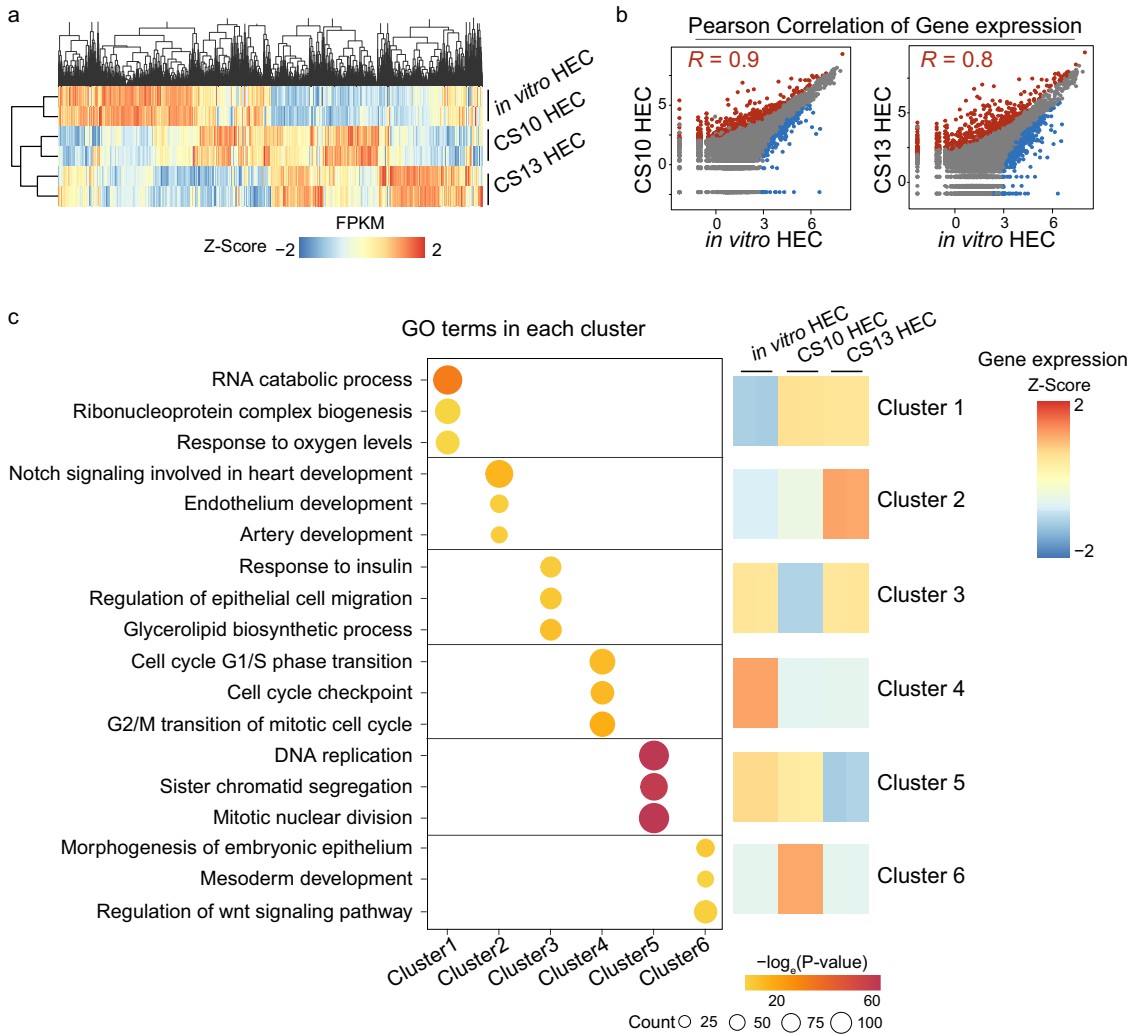

**Fig. 5 Comparison of single-cell transcriptome between in vitro HEC and in vivo HEC. a** Heatmap showing the gene expression patterns in CS10 HEC, CS13 HEC and in vitro HEC. The color gradient represents gene expression value. **b** Scatterplots showing the correlation between in vitro HEC, CS10 HEC, and CS13 HEC, respectively, with the coefficient of determination. The differentially expressed genes are highlighted. Blue, red, and gray dots represent the upregulated genes, downregulated genes, and stable genes in in vitro HEC respectively. **c** GO analysis of top differential upregulated genes in CS10 HEC, CS13 HEC, and in vitro HEC. The dot color shows the enrichment from Fisher's exact *p* value. CS Carnegie stage, HEC hemogenic endothelial cell. Source data are provided as a Source Data file.

research, where the authors found that *EZH1*, a component of a Polycomb repressive complex-2 (PRC2) that mediates gene silencing through histone H3K27 methylation, directly binds to promoters of HSC associated genes, such as *HES1*, *MEIS1*, and *HOX* clusters. *EZH1* deficiency increases arterial- and HSC-associated genes expressions, such as *NOTCH*, *HES1*, and *SOX17*[51]. Another study showed that Retinoid Acid (RA) signaling is required for definitive HSC development in vivo[52]. It is

well known that RA signaling can activate *HOX* genes, and the medial *HOXA* genes promote hESC-derived HSPC maintenance in culture[53]. Therefore, inhibiting the PRC2 complex and fine-tuning RA signaling may help remove the H3K27me3 marks on the above *HOXA* genes to facilitate the generation of HPCs with improved differentiation potentials from hPSCs.

ScRNA-seq analysis revealed that the in vitro HEC generated in our differentiation system is more similar to CS10 HECs than

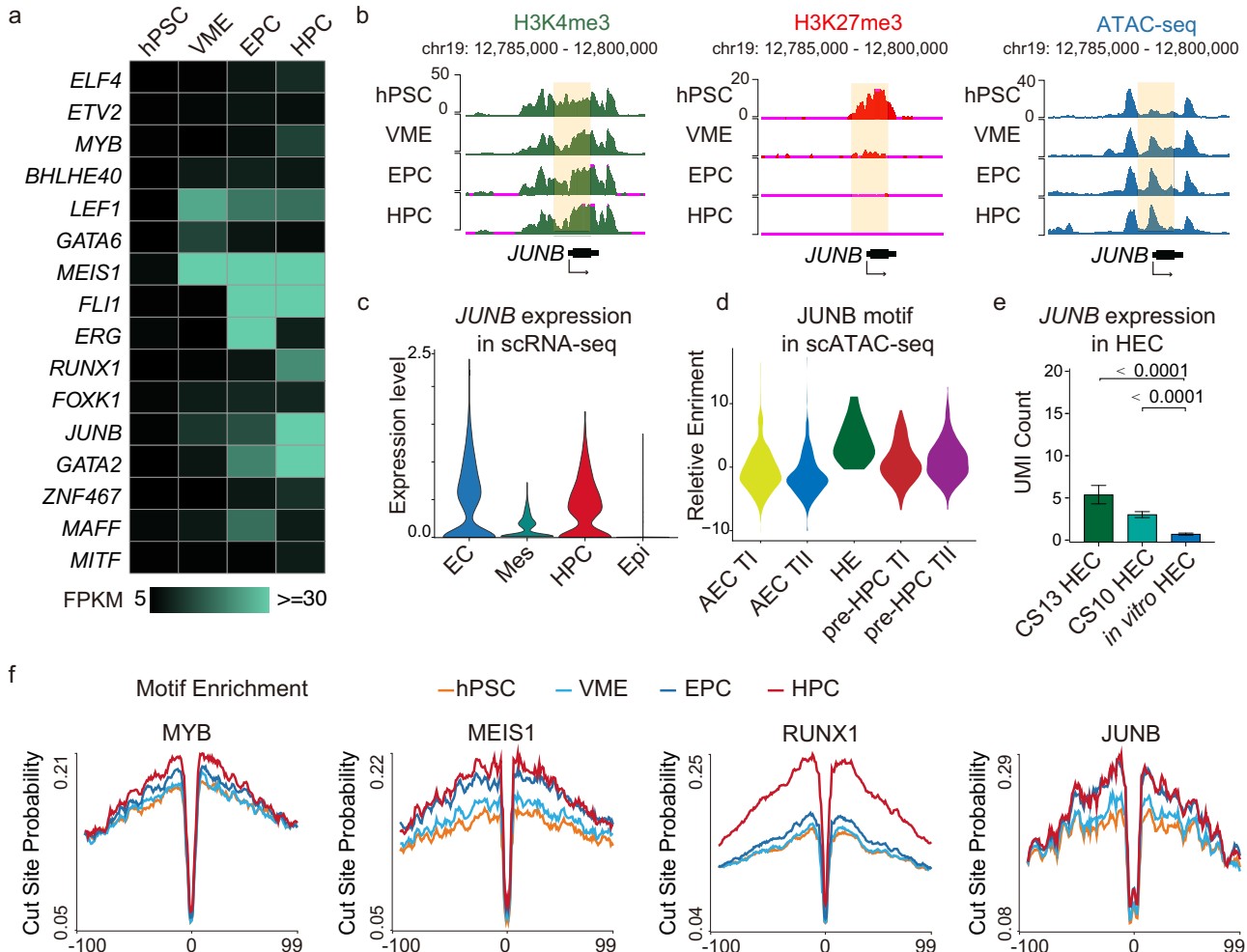

**Fig. 6 Epigenetic profiling identified JUNB as a potential regulator of hematopoietic specification. a** Heatmap showing expression levels (FPKM) of bivalent TFs whose binding motifs are enriched in categories II and III in Fig.2g. The color gradient represents gene expression value. **b** The UCSC genome browser snapshots showing the signal of H3K4me3, H3K27me3, and ATAC-seq peaks at the putative promoter of *JUNB*, and the putative promoters are shaded. **c** Violin plots showing *JUNB* expression is higher in EC and HPC clusters in the scRNA-seq map in Fig. 4b. **d** Motif enrichment of JUNB in the AEC TI, AEC TII, pre-HPC TI, pre-HPC TII, and HE populations. **e** Bar plot showing the JUNB expression levels in CS10 HEC (*n* = 124), CS13 HEC (*n* = 21), and in vitro HEC (*n* = 119). Data are presented as mean values ± SEM. *p* values, two-tailed unpaired Student's *t* test. **f** Visualization of the ATAC-seq footprint for MYB, MEIS1, RUNX1, and JUNB motifs during HPC differentiation. ATAC-seq signals across all these motif binding sites in the genome were aligned on the motif and averaged. hPSC human pluripotent stem cell, VME vascular mesoderm cell, EPC endothelial progenitor cell, HPC hematopoietic progenitor cell, Epi epithelial cell, Mes mesoderm cell, EC endothelial cell, HE hemogenic endothelium, AEC arterial endothelial cell, AEC TI AEC-type I, AEC TII AEC-type II, pre-HPC TI pre-HPC type I, pre-HPC TII pre-HPC type II. Source data are provided as a Source Data file.

CS13 HECs. Notably, the arterial genes are lower, and cell cycle genes are higher in in vitro HEC than in CS13 HECs. During mammalian embryo development, the definitive hematopoiesis is initiated from arterial HECs in AGM[8,54]. The HECs generated by using our protocol also have some arterial characteristics and highly expressed arterial endothelium genes, such as *GJA4* and *JAG1* (Fig .5f). This is consistent with previous studies that the growth factors or small molecules such as VEGF, FGF2, and SB431542 can promote AEC specification[55]. However, the arterial features of in vitro HECs are still significantly weaker compared to CS13 HECs. Insulin plays a crucial role in glucose and lipid metabolism[56] and is supplemented in most serum-free differentiation mediums. Interestingly, insulin was reported to suppress the arterial marker gene expression[55]. Insulin withdrawal significantly increases arterial ECs formation at the cost of HPC[9,19]. Therefore, optimizing insulin concentration and cell cycle adjustment may help produce arterial HECs with lymphoid differentiation and definitive hematopoiesis potential.

We identified JUNB as a regulator of in vitro HPC differentiation. JUNB belongs to the basic leucine zipper (bZIP)/AP-1 TF family, which can activate target gene expression in response to signals from growth factors and cytokines[57,58]. *JUNB* KO mouse has severely impaired yolk sac blood vessel development. Inhibiting AP-1 TF functions using a dominant-negative FOS peptide in mouse ESCs causes a shift in the balance between smooth muscle and hematopoietic fate[57]. *JUNB* is also a target gene of hypoxia-induced signaling and was required for VEGF transcription in response to hypoxia[59]. In this study, *JUNB* knockout did not impair the generation of CD34+ EPCs or CD31+ ECs (Supplementary Fig. 6g, h), which may be explained by the VEGF supplement in the differentiation medium, which compensated for reduced endogenous VEGF activation in *JUNB* KO cells. The generation of HEC and HPC is specifically inhibited by the lack of JUNB (Fig. 7b, d). At the chromatin level, JUNB binds to the promoter of several key hematopoiesis TFs such as RUNX1, and the expression levels of many hematopoiesis

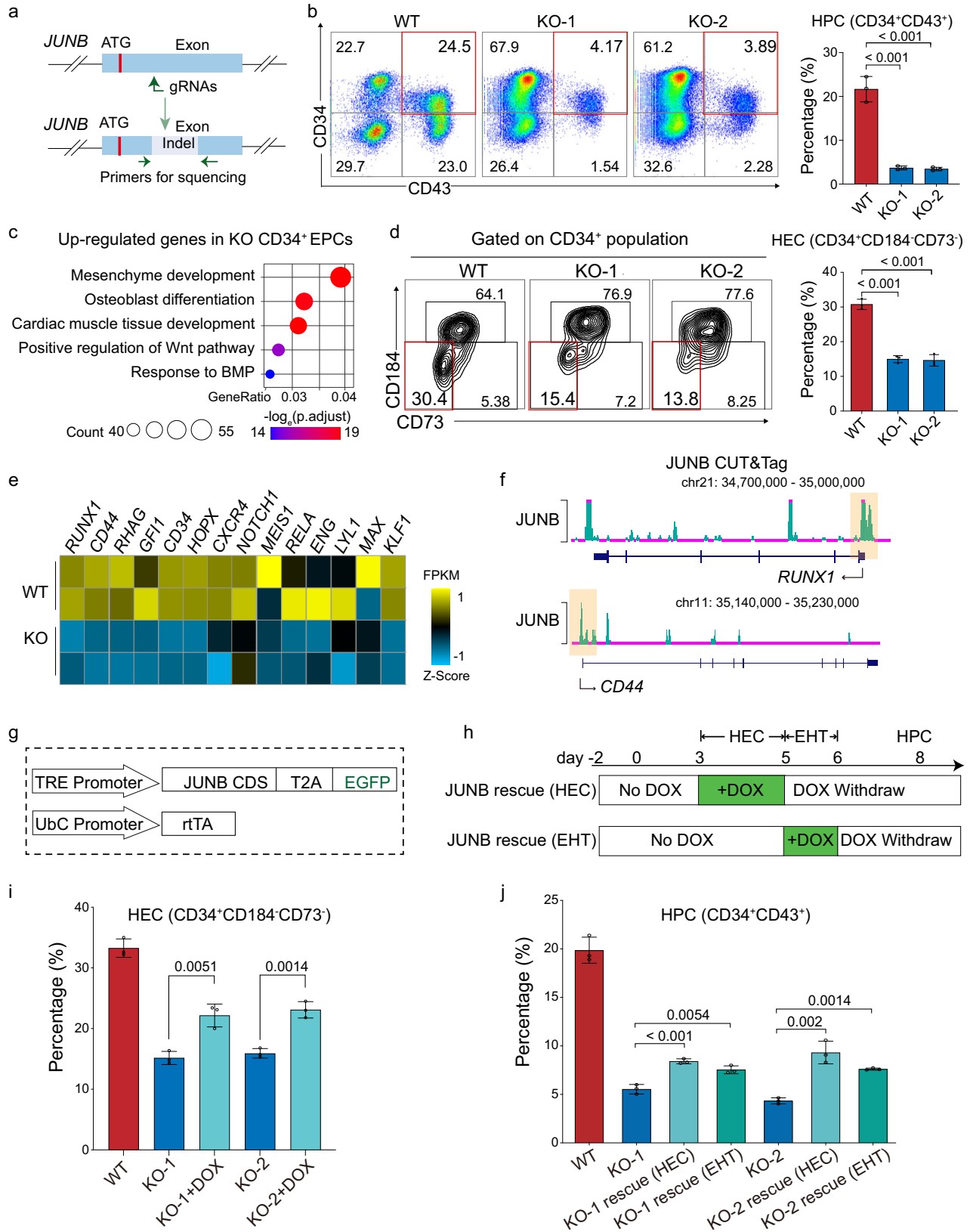

regulators are significantly lower in *JUNB* KO day 5 CD34+ EPCs. Furthermore, 34% of the open chromatin peaks are attenuated in *JUNB* KO HECs. Rescue experiments showed that JUNB's action window encompasses HEC formation and EHT (Fig.7i, j). JUNB overexpression (JUNB OE) alone does not cause any significant increase in the percentage of HECs and HPCs

(Supplementary Fig. 7g, h, i). It seems that during HPC differentiation from hPSC, JUNB is specifically needed for the robust emergence of HECs and HPCs. Interestingly, several studies suggested that the AP-1 family of TFs can serve as pioneer factors to open up chromatin regions so that lineage-specific TFs can bind to their target genes and establish cell identities[60,61]. Thus, in

**Fig. 7 JUNB deficiency severely impaired hPSC differentiation to HECs and HPCs. a** Schematic showing sgRNA targeting *JUNB* in proximity to the start codon. Green arrows represent genotyping PCR primers. **b** Representative flow cytometry density plots and quantification of HPC (CD34$^+$ CD43$^+$) on day 8 in WT and KO cells, $n = 3$ independent experiments. **c** GO enrichment of genes upregulated in the *JUNB* KO CD34$^+$ EPCs compared to WT. $p$ values, Fisher's exact test. **d** Representative flow cytometry density plots and quantification of HEC (CD34$^+$ CD184$^-$ CD73$^-$) populations on day 6 in WT and *JUNB* KO cells. $n = 3$ independent experiments. **e** Heatmap showing the expression level of genes important for HEC identity in WT and *JUNB* KO HECs. The color gradient represents gene expression value. **f** TF binding for JUNB at RUNX1 and CD44 genomic loci. The putative promoters are shaded. **g** The illustration of *JUNB* inducible expression constructs. **h** The schematic showing the DOX induction window during HPC differentiation. **i** Bar plot showing that DOX-induced ectopic *JUNB* expression from day 3 to day 5 partially rescued HEC (CD34$^+$ CD73$^-$ CD184$^-$) percentage. $n = 3$ independent experiments. **j** Bar plot showing that induction of *JUNB* expression with DOX during either HEC specification or EHT window rescued HPC (CD34$^+$ CD43$^+$) percentage significantly. $n = 3$ independent experiments. HPC hematopoietic progenitor cell, HEC hemogenic endothelial cell, EHT endothelial-to-hematopoietic transition, WT wild type, KO knock out. For bar graphs **b**, **d**, **i**, **j**, data are presented as mean values ± SD. $p$ values, two-tailed unpaired Student's $t$ test. Source data are provided as a Source Data file.

HECs, JUNB may act as a pioneer factor for EHT, making the neighboring regions of its recognition sites accessible for master hematopoietic TFs.

It is likely that JUNB is also an essential regulator of HEC and HSC formation in the early human embryo. ScRNA-seq analyses of CS10 and CS13 HECs reveal that *JUNB* is expressed at higher levels in in vivo HECs than in vitro HECs (Fig. 6e). Data mining from another scRNA-seq study reveals that *JUNB* is also highly expressed in in vivo HSPC compared to HSPCs generated from hPSC[62]. Thus, *JUNB* is expressed at the right place and time during early human embryo hematopoiesis. Loss of function study revealed that JUNB could impact the expression and chromatin binding of many hematopoietic regulators during the HEC formation and EHT window. Therefore, JUNB might also be involved in in vivo HEC and HSC formation in the human embryo.

In summary, our comprehensive epigenetic and transcriptional profiling and integrative analysis demonstrate that under highly defined in vitro HPC differentiation conditions, chromatin changes pave the way for the sequential emergence of progenitor cell types, leading to HPCs formation resembling the in vivo hematopoiesis in the human embryo. Moreover, functional studies motivated by these analyses uncovered that JUNB deficiency significantly inhibits HEC and HPC generation in the in vitro system. The current study enriched the knowledge about the epigenetic regulation of hematopoietic specification in the human system and may provide clues to optimize in vitro differentiation and obtain HECs and HPCs with more robust hemogenic potentials.

## Methods

**Statement**. All human embryonic stem cell studies were approved by the Institutional Review Board of Tsinghua University. This article does not contain any studies with human subjects performed by any of the authors.

**Human pluripotent stem cell culture and differentiation**. H1 hESC line (WiCell Institute) and iCRISPR/Cas9 cell line (generous gift from Dr Huangfu, Sloan-Kettering Institute, New York, USA) were used in this study. HPSCs were cultured on MEF feeders in the hESC medium containing KnockOut DMEM (Gibco) culture medium supplemented with 20% (vol/vol) KnockOut serum replacement (Gibco), 1% nonessential amino acids (Gibco), 1 mM L-GlutaMAX-I (Gibco), 0.1 mM β-mercaptoethanol (Sigma-Aldrich), and 8 ng/ml bFGF. They were passaged with 1 mg/ml collagenase IV (Invitrogen) and seeded onto inactivated feeders grown on 0.1% gelatin (Sigma Aldrich) pre-coated plates. For differentiation, hESCs were maintained on Matrigel (BD Biosciences)-coated plates (Corning) in TeSR-E8 medium.

To induce hematopoietic differentiation, undifferentiated hPSCs cultured in TeSR-E8 medium were dissociated into single-cell suspension with Accutase (Millipore) and plated onto Matrigel-coated culture dishes at a density of $2 \times 10^4$ cells/cm$^2$ in TeSR-E8 medium with Y27632 (TOCRIS) (10 μM). After 24 h, cells were induced by first culturing in RPMI1640 medium supplemented with 2% B27 (Gibco, with or without insulin), 1% L-GlutaMAX-I, 50 μg/ml Ascorbic acid (Sigma Aldrich). Five ng/ml BMP4 (R&D) was added for 24 h. Afterward, the medium was changed to RPMI1640 medium containing 5 ng/ml BMP4 and 2 μM GSK3 inhibitor CHIR99021 (TOCRIS) for another 48 h. In differentiation stage 2,

cells were re-plated onto Matrigel-coated dishes at a density of $4 \times 10^4$ cells/cm$^2$ in basal medium with 50 ng/ml VEGF165 (11066-HNAH, Sino Biological) and 10 ng ng/ml FGF2 (10014-HNAE, Sino Biological) for 48 h. The medium was replaced with RPMI1640 medium supplemented with 50 ng/ml VEGF165, 10 ng/ml FGF2, and TGFβ inhibitor SB431542 (TargetMol, 10 μM) for another 72 h. All the cells used in this study were verified without mycoplasma contamination.

**Flow cytometry analysis**. Cells were dissociated into single-cell suspension with accutase (Millipore) or rinsed off from the plate wherever suitable and resuspended in a FACS washing buffer (PBS with 5% fetal calf serum (FCS) and 2.5 mM EDTA). The cell suspension was stained with desired antibodies for 30 min at 4 °C. Then cells were washed and suspended in 0.5% bovine serum albumin in phosphate-buffered saline (BSA/PBS) buffer. Data were collected with FACS Caliber flow cytometer (BD) and analyzed using FlowJo software (version 10.4). Cells were gated as shown in Supplementary Fig. 6e–h and Supplementary Fig. 7a, c. Antibodies used in this study were described in supplementary Table 1.

**Bulk RNA-seq library preparation**. Total RNA was extracted from the freshly sorted cells with RNeasy Plus Mini Kit (Qiagen) according to the manufacturer's instructions and treated with RNase-free DNase. RNA-seq libraries were prepared using Smart-seq2[63] as manufacturer's instructions[63]. In brief, 50 ng purified total RNA were transferred to a tube containing 1.9 μl of 0.2% triton-100, 0.1 μl RNase inhibitor (40 U/μl),1 μl of oligo-dT primer (10 μM) and 1 μl of dNTP mix (R0192, Fermentas), and incubated at 72 °C for 3 min to remove the RNA secondary structure. After the reverse transcription reaction, cDNA was purified using Agencourt AMPure XP beads (Beckman Coulter). The sequencing libraries were prepared using TruePrep DNA Library Prep Kit V2 for Illumina (TD502-01, Vazyme). The sequencing was done on Hiseq X Ten (Illumina).

**ATAC-seq library preparation**. ATAC-seq libraries were prepared following the standard ATAC-seq protocol with minor modifications[64]. Briefly, 50,000 sorted cells were lysed on ice in a tube containing 50 μl of cold lysis buffer, the nuclei were collected and subjected to transposition reaction in a preheated metal bath at 37 °C. Then the DNA fragments were extracted and purified by a PCR purification kit. The DNA was then resuspended in 29 μl of ddH$_2$O and used in the library construction process according to the manufacturer's protocol (TD202 and TD502, Vazyme). Sequencing libraries were prepared according to the manufacturer's protocol. The DNA libraries were sequenced on the HiSeq X Ten platform.

**ChIP-seq library preparation**. ChIP-seq library was prepared according to the ultra-low-input micrococcal nuclease-based native ChIP (ULI-NChIP) procedure[65]. Briefly, 50,000 sorted cells were used per reaction. The sorted cells were washed two times in 0.5% BSA/PBS solution. The nuclei were collected and subjected to fragmentation reaction in a preheated metal bath at 37 °C for 5 min. After immunoprecipitation reaction with desired antibody, DNA fragments were extracted using Phenol-Chloroform. One microgram of either H3K27me3 antibody (pAb-069-050, Diagnode) or H3K4me3 antibody (9727, Cell signaling Technology) was used for each reaction. Antibodies used were described in supplementary Table 1. Sequencing libraries were prepared according to KAPA Hyper Prep Kit with minor modifications. All the generated DNA libraries were sequenced on the HiSeq X Ten platform.

**CUT&Tag library preparation**. CUT&Tag was performed according to the instruction of the NovoNGS® CUT&Tag 2.0 kit (N259-YH01, Novoprotein). Briefly, 40,000 sorted cells were immobilized on Concanavalin A magnetic beads and permeabilized. Cells-beads complex were incubated with the primary antibody the secondary antibody. Unbound antibodies were then removed from cells-beads complex after twice washing. The nuclei were collected and subjected to transposition reaction in a preheated metal bath at 37 °C. Then the DNA fragments were extracted, purified and subjected to library construction using the reagents

provided by the kit. Antibodies used in this study for CUT&Tag are JUNB (3753 S, 1:50, Cell Signaling Technology), guinea pig anti-rabbit IgG (SAA544Rb50, 1:100, Cloud clone).

**Single-cell RNA-seq library preparation**. Differentiated cells on day 6 were digested and filtered through a 70 μm cell strainer to get a single-cell suspension. Live/dead assay showed that around 90% of the cells were viable. Then, single-cell suspensions were immediately loaded on Chromium Single Cell Controller (10× Genomics) for droplet formation, and cDNA and subsequent sequencing libraries were generated using Single Cell 3' Reagent Kit (10× genomics) according to the manufacturer's protocol. All the generated libraries were sequenced using Illumina sequencer with paired-end 150 bp (PE150) reading strategy (CapitalBio Technology, Beijing).

**Bulk RNA-seq data analysis**. Reads were trimmed with the Trim_Galore (version 0.6.7) software with the default parameter. Then, Adaptor-trimmed paired-end RNA-Seq reads were mapped to hg38 whole genome using STAR (version 2.7.1a) with default parameters. The read count was calculated using htseq-count (version 0.11.1) with parameters -r pos -a 10 -t exon -s no -i gene_id -m union. The raw data count normalization and differentially expressed genes were obtained using DESeq2 (version 1.32.0) software.

**ChIP-seq and ATAC-seq data processing**. All of the paired-end raw reads were processed using Trim_Galore (version 0.6.7) to trim adaptor and low-quality reads. Adaptor-trimmed paired-end reads from ATAC-seq and ChIP-seq were aligned to hg38 by bowtie2 (version 2.3.5)[66] with the parameters:–t–q–N 1–L 25. The SAM were converted to BAM file using samtools (version 1.9) software. PCR duplicates and multiple mapped reads were removed with Picard tools (version 2.26.10, MarkDuplicates). The replicate BAM files for each time point were merged and removed reads aligning to blacklisted genomic regions before calling peaks with MACS2[67] (version 2.2.6), and the enriched TF motifs were analyzed by Homer (version 4.10.4, findMotifsGenome). Signal tracks for each sample were generated using the bamCoverage (version 3.3.0, deeptools) function and were normalized to RPKM for visualization.

**ATAC-seq cluster analysis with Mfuzz**. After the ATAC-seq peak for each time point was merged within 100 bp using bedtools (version 2.27.1) merge function, a consensus union set of peaks was created by merging peak sets from all stages. Subsequently, ATAC-seq read counts for each sample were calculated using the bedtools multicov function default settings. Based on ATAC-seq read counts, RPKM values were calculated. The RPKM matrix was subjected to quantile normalization with RPKM value >5 at least two time points. Before clustering, Peaks with RPKM value with a coefficient of variation (CV) across time <10% were removed to preserve the dynamically changing peak regions. The remaining peaks were quantile normalized, and Z-score transformed and then subjected to mfuzz[68] (version 2.56.0) clustering in R to show the dynamic chromatin open state during HPC differentiation.

**ChromHMM analysis and identification of bivalent genes**. ChromHMM (version 1.22) software[35] was adopted to identify and characterize chromatin states as instructed in the manual. The alignment file of H3K4me3 and H3K27me3 from hPSC were binned into 200 bp bins using the BinarizeBam command. LearnModel was used with four emission states using default parameters. Four chromatin states were classified: H3K27m3-only, H3K4me3-only, H3K27me3/H3K4me3 (termed bivalent domain), and none (non-marked region). The bivalent genes were defined as follows: firstly, the bivalent regions were extracted from the segmentation file of hPSC, then the promoter region (defined as ± 2.5 kb around the TSS) of RefSeq transcripts was intersected with bivalent regions. Only those with overlapping lengths >200 bp were counted as bivalent genes.

**Gene ontology analysis**. The R package clusterProfiler (version 3.6.0) was used to perform Gene Ontology (GO) enrichment with default arguments. GO terms for each cluster were re-summarized and visualized as bar plots using a customized script, and Fisher's exact p value were plotted to show the significance. GO terms with significance and genes in each GO terms were listed in the Supplementary Data file.

**Single-cell RNA-seq data process and analysis**. The quality of raw sequencing data was checked using FastQC (version 0.11.9), and single-cell expression matrices were generated following the cellranger count pipeline (version 2.0.1). For integrated analysis of multiple samples, the cellranger aggr pipeline was used to normalize samples to equal read depth. The resulting UMI (unique molecular identifier) matrices from the cellranger pipeline were used in the following analysis. To check differentiation efficiency, we performed differentiation using H1 and iPS cell lines and mixed cell culture after SB431542 treatment for single-cell sequencing. We then used Souporcell (version 2.0) to distinguish source cell lines in the data using SNP information and retained cells from H1 for downstream analysis. For cell clustering and the identification of differentially expressed genes, cell type identification was done using the R package Seurat (version 3.2). Briefly, cells were

filtered using 3 criteria: number of expressed genes, number of UMI count, and percentage of mitochondrial genes in each cell. Cells with any of these metrics fall out of 2 standard deviations of all cells are discarded. Next, highly variable genes were identified, and these genes were used to perform principal component analysis. Ten principal components were used to perform cell clustering using the FindClusters function in Seurat with the resolution parameter set to 0.15. Differentially expressed genes were identified using the FindAllMarkers function, comparing each cell cluster against the rest of all cells. Top DEGs were used to generate heatmaps showing marker gene expression for cell clusters.

**Establishment of *JUNB* knockout cell line**. SgRNA targeting *JUNB* was designed using the online tool (http://chopchop.cbu.uib.no/). The sgRNA sequences and primers for genotyping are listed in supplementary Tables 2 and 3, respectively. SgRNA expression lentiviral vector was constructed by inserting the annealed 23-bp protospacer oligonucleotides (100 μM per oligo) into the pLKO5.sgRNA.EFS.GFP (Addgene: 57822) vector linearized with BsmBI. 293 T cells were transfected with the sgRNA plasmid to produce lentivirus.

The iCas9 hPSCs were used to generate knockout cell lines in this study. Before sgRNA transfection, iCas9 hPSCs were treated with 1 μg/ml of doxycycline (D9891, Sigma) and 10 μM of Y-27632 for 24–48 h. The following day, cells were digested to single-cell suspension using accutase (Millipore) and then seeded on MEF feeders coated plate containing 1 μg/ml of doxycycline, 10 μM of Y-27632. Once the cells were attached, the sgRNA lentivirus was added to the culture dish for 8 h. Two days after transfection, hPSCs were dissociated into single cells by accutase and re-plated at ~1000 cells per 6 cm dish with 10 μM of Y-27632. Cells were allowed to grow until colonies from single cells became visible (~10 days). Single colonies were picked and analyzed by Sanger sequencing to identify *JUNB* KO clones.

**Western blotting**. Total cell samples were solubilized in 1 × SDS buffer, boiled in a heat block at 95 °C for 10 min and loaded in SDS-PAGE with 1 × 10⁵ cells per lane. Following electrophoresis, proteins were transferred to Nitrocellulose Transfer Membrane (#66485, BioTrace). The membranes were then blocked with 5% BSA for 1 h at room temperature and incubated with primary antibodies overnight at 4 °C (anti-JunB: 1:1,000 dilution, Abcam; anti-Histone H3: 1:10,000 dilution, ABclonal). The membranes were later incubated with HRP-conjugated goat anti-rabbit IgG antibodies (1:5,000 dilution, 111-035-144, Jackson ImmunoResearch) for 30 min at room temperature. After washing, the membranes were developed and captured by ECL reagent (WBKLS0100, Merck Millipore) and Bio-Rad ChemiDoc Imaging System. Antibodies used were described in supplementary Table 1. Images of uncropped blots with molecular weight markers are included in the Source Data file.

**Quantitative PCR**. Total RNA was extracted from fresh cells using TRIzol according to the manufacturer's instructions. Two μg total RNA was reverse transcribed with 5× All-In-One RT MasterMix. Q-PCR were performed in quadruplicate using GoTaq qPCR Master Mix with specific primers for each gene. The sample input was normalized against the Ct (Critical threshold) value of GAPDH. Primer sequences are listed in supplementary Table S4.

**Statistical analysis**. Quantitative data are shown as mean ± standard deviation (SD). The statistical significance was determined using two-sided Student's *t*-test using Prizm 9 (version 9.2.0 (283)) software. No adjustments were made for multiple comparisons. The number of sample sizes is noted in the figure legends.

**Reporting summary**. Further information on research design is available in the Nature Research Reporting Summary linked to this article.

## Data availability
The high-throughput sequencing data and single cell data generated in this study have been deposited in the GEO repository database under accession number GSE168372. Data that support the findings of the current study have been included within the paper and the Supplementary Information file. Source data are provided with this paper. All other data supporting the findings of this study are available from the corresponding author on reasonable request.

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

## Acknowledgements

We thank Dr. Kai Xu (Peking University, China) and Dr. Fengzhi Zhang (Beijing Huaxin Hospital, China) for constructive suggestions. We thank Dr. Wen Wang (Tongji University, China) for the help in data analysis. This work was supported by the National Key R&D Program of China Grant (2019YFA0110001 and 2017YFA0102802), the National Natural Science Foundation of China (NSFC) Grant (31970819, 91740115, 31771108) to J.N., 32000610 to J.G. The Tsinghua University Spring Breeze Fund (2021Z99CFY033) to J.N.

## Author contributions

X.C. and J.N. conceived and designed the experiments. X.C. performed cell differentiation, ATAC-seq, ChIP-seq, RNA-seq library preparation, and bioinformatics analysis. X.C. and H.Q. analyzed scRNA-seq and scATAC-seq data. PW performed functional characterization of *JUNB* KO cells. Y.Zhu, Y. Zhang, X.Z., F.D., S.D., J.G., Y.H. contributed to cell differentiation, characterization, and data assembly. X.C. and J.N. wrote the manuscript. All authors approved the manuscript.

## Competing interests

The authors declare no competing interests.
