## [Peer Review File · Nature Communications]

Integrative epigenomic and transcriptomic analysis reveals the requirement of JUNB for hematopoietic fate inductionREVIEWER COMMENTS

Reviewer #1 (Remarks to the Author):

The manuscript by Chen et al. describes the molecular characterization of haematopoietic specification of human pluripotent stem cells (hPSCs). Despite several studies have described the development of haematopoietic lineages from hPSCs, the detailed mechanisms regulating the emergence of haematopoietic cells have not been elucidated. This translates in an overall poor efficiency of hPSC haematopoietic differentiation. Therefore, the question addressed is timely and of interest, as a better characterization of haematopoietic specification is needed and could, in principle, be exploited for the generation of transplantable haematopoietic cells from hPSC as well as for in vitro disease modelling. The results are interesting, however in my opinion there are some limitations that are left unanswered and need to be addressed to grant a publication in Nature Communication.

Specific points:

This reviewer understands the choice of focusing on bivalent genes. However, bivalence seems a poor predictor of a cell state when compared to ATAC-seq. Many genes, including JUNB which is the focus of the last part of the paper, is bivalent already in hPSCs a totally irrelevant stage for hematopoietic specification, but it is not expressed until the endothelial stage. Given this, this reviewer finds the analysis of bivalent genes in hPSCs that show a dramatic activation or repression in HSPCs not extremely relevant or meaningful. Since the authors have collected already all the data, it would be much better to analyze at differences between the different steps, so to map the developmental changes that occurs throughout the hematopoietic specification. In other words, what changes between the hPSC and mesoderm stage? Between mesoderm and endothelial cells? Between endothelial cells and HSPCs? Focusing the comparison between successive cell states will very likely uncover more relevant regulators and be much more helpful to the community.

The authors made the effort to compare in vitro derived hemogenic endothelial cells (HECs) with those found in the embryo. However, they only cherry-picked a particular stage of human embryonic development, CS13. This seems to be unfair, as the authors are clearly aware that there are other HECs which are thought to be devoid of HSC potential. As such, the authors should reperform similarity analysis of their cells comparing them to both CS10 and CS13 HECs. As their hPSC-derived HECs do not express HOXA genes, these cells are likely reflecting extra-embryonic progenitors, which are less capable to generate lymphoid cells and HSC.

As a reference, they should also compare in vitro-derived arterial cells with those found at CS10 and CS13 as well.

In addition, can the authors generate HOXA+ HECs or HECs with lymphoid potential so to verify that what they have described in the current manuscript are general principles of hematopoietic specification and is not restricted to a HOXA- developmental program?

The fact that hematopoietic development is dependent on activation of AP-1 TF family is already known. In Obier et al (Development 2016), the Bonifer group have already described part of the downstream effectors of the JUN axis during hematopoietic development, using a different strategy. This paper should be referenced and commented. In addition, since what is downstream of JUN is not exactly novel, can the authors use their thorough database to identify what triggers JUN activity (EGF, TNF or other cytokines? Hypoxia?) This would be novel and very useful for the wide community of laboratories differentiating hPSCs in blood cells.

When exactly JUNB plays a role in hematopoietic specification of hPSCs? No hematopoietic lineages are generated from JUNB KO but is unclear whether this is because HECs are absent and/or unable to make the transition to blood cells. Can the authors perform rescue experiments, overexpressing JUNB at the two critical stages (HEC specification and EHT) to see when it is required?

Minor points:

- The authors claim that CD44 expression is regulated directly by JUNB. But CD44 is also highly expressed in arterial cells and the CD184+ fraction representing cells with an arterial fate are present in JUNB KO differentiating cells. Is CD44 expression absent in the CD184+ cells as well or the lack of CD44 expression in JUNB KO cells is specific to HECs?
- Since HSCs are not generated via the protocol used in these studies, remove "S" from HSPC and refer to those cells as HPCs.
- KDR is the correct gene symbol for FLK1
- line 290: HAEC are human and not hemogenic arterial endothelial cells.
- There are several typos and language issues in the manuscript. Please proofread carefully to correct these, taking care of homogenizing the use of past and present tenses throughout the manuscript.

Reviewer #2 (Remarks to the Author):

The paper by Chen et al. describes epigenomic and transcriptomic analysis of cell populations emerging during hematopoietic differentiation of H1 hESCs. By analyzing hESC bivalent genes which get active during hematopoietic differentiation, authors discovered that JUNB has a bivalent promoter in hESCs and get activated in endothelial and hematopoietic cells. To find out whether JUNB has effect on hematopoietic differentiation, JUNB knockout hESCs were generated. These knockout cells failed to produce blood. By identifying JUNB as a master regulator of hematopoietic commitment in hESC differentiation culture this paper makes a novel contribution to our understanding of transcriptional program regulating hematopoietic development.

Comments:

1. To increase confidence in the obtained JUNB results and eliminate a possibility of off-target effects, authors should demonstrate if similar results can be obtained using several JUNB knockout clones. In addition, rescue experiments should be performed to show a restoration of hematopoietic potential in JUNB knockout cells following introducing exogenous JUNB.
2. What type of hematopoiesis produced in this system, extraembryonic or intraembryonic? Does JUNB affect intraembryonic or extraembryonic-type hematopoiesis or both? What types of CFU this protocol produces? Do CD34+ cells generated in this protocol possess lymphoid potential?
3. Please describe experimental design for experiments depicted in Fig.4. What was the starting population for these experiments, isolated CD34+ cells?
4. How hemogenic endothelial cluster and HPC clusters were identified? What are the differences in HPC-T1 and T2 clusters? Please provide in supplement RNAseq UMAP plots with marked RUNX1, CD44, SOX17, CDH5 and CD34 expression.
5. Authors found that hemogenic endothelium generated in hPSC cultures is highly proliferative. What about CS13-HEC?
6. In introduction, authors describe just two waves of embryonic hematopoiesis and failed to acknowledge its complexity and multiple waves (see DOI: 10.1038/nrm.2016.127).
7. Authors claim that JUNB knockout did not impair the generation of CD34+ EPCs. However, CD34 is broadly expressed in non-endothelial cell types. To ensure that this statement is correct, additional endothelial markers, such as VE-cadherin and CD31 should be evaluated in WT and KO cultures.

Minor:

1. Ref 5 and 6 are related to EHT in AGM region and are not related to EHT during primitive hematopoiesis.
2. Ref. 7 is incorrect. This reference describes the effect of VEGF and FGF2 on HUVECs and has nothing to do with mesodermal differentiation.
3. Line 50: hematopoietic endothelium should be hemogenic endothelium.
4. In result section, please introduce hPSC line used in this study (H1 hESC).
5. Line 140. SOX17 is involved in EHT, the major function of this gene is to promote arterial commitment.
6. Line 168. Correct H3K37me3 typo.
7. Line 273. FLK1 differences are negligible and not significant. Word "noticeable" should not be used. Please use the current KDR nomenclature for FLK1.

Reviewer #3 (Remarks to the Author):

To understand the mechanism of HSPC fate determination in humans, the authors dissect the epigenomic roadmap from hPSCs to HSPCs by profiling chromatin accessibility, histone modifications and transcriptome. Generally, the epigenetic feature dynamics and gene expression dynamics are highly correlated during differentiation. For the chromatin accessibility, the regulatory regions become accessible before key TF binding to the chromatin. For the histone modifications, the bivalent genes are characterized by stage-specific H3K4me3 and H3K27me3 during HSPC differentiation. Specifically, they reveal that EHT contains several intermediate subpopulations with unique transcriptome and chromatin states. Furthermore, they identify JUNB as a new regulator of HSPC differentiation and the deficiency of JUNB by iCRISPR will impair HEC formation and EHT.

Major comments:

1. Whether the differentiation protocol used in this study can generate functional HSPCs with complete self-renewal and engraftment abilities remains unknown.
2. Related to comment #1, if the generation of HSPC with complete engraftment ability is difficult to achieve, whether the profiling of epigenetic features and transcriptome features in this study can resolve the bottleneck of induction of real HSPC in vitro.
3. Sc-RNAseq data showed that Junb is expressed in EC and HPC populations. Functional analysis of Junb showed that it could regulate hematopoietic specification and ChIP-seq data showed hematopoietic genes were direct targets of JUNB. However, how JUNB regulates hematopoietic TFs specifically remains unclear.
4. JUNB deficiency impaired HEC and HSPC differentiation in vitro. Whether it can play the similar role in vivo? Can overexpression of JUNB facilitate the generation of functional human HSCs in vitro?

Minor comments:

1. The y-axis of the Fig 4h is not labeled.
2. The result of Fig 4g shows the developmental path of EHT, have you tried other analysis methods, such as RNA velocity to validate this result?
3. Since a lot of sequencing omics data have been obtained, why not building a website to display all the omics data in a visual way, so that readers can better use this information?

We thank the careful reading and critical comments from three reviewers.
Those comments are valuable and very helpful for improving our manuscript.
We have now conducted more experiments and extensive new analyses to
address all reviewers' concerns. Please see the point-to-point responses below.
The revised texts are marked in blue in the manuscript. To avoid confusion, we
used Fig. 1, 2, 3, etc., to refer to Figures in the revised manuscript and Fig. R1,
R2, R3, etc., to refer to Figures in this response letter.

**General comment 1**

**Both reviewer 1 (specific point 2nd paragraph) and reviewer 2 (comment 5)**
**think that we should compare *in vitro* HECs to *in vivo* CS10 and CS13**
**HECs.**

**Response:** We thank the reviewers for the valuable suggestion. Following the
reviewer's advice, we compared *in vitro* HECs to CS10 HECs and CS13 HECs
in parallel. **We have added the following results in the revised manuscript**
**(Fig. 5, Lines 238-255, Page 10-11).**

The previous study showed two temporally and molecularly distinct HEC
populations in the developing human embryo, and they first appear at CS10
and CS13, respectively. Furthermore, CS10 HECs are thought to be devoid of
definitive HSC potential, while CS13 HECs are considered to be HSC-primed
HECs¹. *In vitro* differentiated HPCs often have low lymphoid differentiation
potential and are biased towards the myeloid lineage. Therefore, we reasoned
that comparing *in vitro* HECs and CS13 HECs would provide logical clues to
improve the differentiation system to obtain HPCs with expanded differentiation
potential.

The results show that CS10 and CS13 HECs cluster together (Fig. R1 a), and
the *in vitro* HECs are more similar to CS10 HECs (Pearson correlation R=0.9)
than CS13 HECs (Pearson correlation R=0.8) at transcriptome level (Fig. R1
a,b). We performed k-means clustering analysis of differentially expressed
genes between *in vitro* HECs, CS10, and CS13 HECs and obtained 6 clusters
(Fig. R1c). Genes in cluster 1 are upregulated in both CS10 HECs and CS13
HECs, and these genes are related to hypoxia, which is attributed to the hypoxic
tissue environment *in vivo* (Fig. R1c). Cluster 2 genes are upregulated
specifically in CS13 HECs, and these genes are mainly involved in endothelium
development and artery development. While *in vitro* HECs and CS10 HECs
show less arterial endothelial features but upregulated genes related to cell
cycle transition (Cluster 4) and DNA replication (Cluster 5), indicating that they
are in an active proliferation state compared to CS13 HECs (Fig. R1c).

The correlation analysis suggest that our *in vitro* HECs are more similar with
CS10 HECs, which are mostly extra-embryonic progenitors and less capable

to generate lymphoid cells and definitive HSCs. Our GO analyses also
 suggest that culture in low oxygen condition, promoting arterial endothelial
 features and cell cycle adjustment, may enhance the potential of *in vitro*
 differentiated HECs to form definitive hematopoietic stem cells (HSCs).

**Fig. R1. Comparison of single-cell transcriptome between *in vitro* HECs**
 **and *in vivo* HEC.**

**a.** Heatmap showing the gene expression patterns in CS10 HECs, CS13 HEC, and *in vitro*
 HEC. **b.** Scatterplots show the correlation between different paired groups with the
 coefficient of determination (R). The differentially expressed genes (defined by FDR < 0.05
 and fold change > 2 with Deseq2) are highlighted. **c.** GO analysis of top differentially
 upregulated genes in CS10 HECs, CS13 HEC, and *in vitro* HEC.

**General comment 2**

Both reviewer 1 (specific point 4th paragraph) and reviewer 2 (comment 1)
 suggest that we use multiple JUNB KO clones and perform the rescue
 experiment of JUNB to prove the specificity of the JUNB KO phenotype.

**Response:** We thank the reviewers for raising this important question. We
 generated another JUNB knock-out hPSC line (Fig. R2a, b) and repeated the
 HPC differentiation experiments using two independent JUNB KO hPSC lines.
 The new experiments show that JUNB deficiency severely impairs HEC and
 HPC generation, which confirms previous conclusions (Fig. R2c, d). We have
 added these results in the revised manuscript (**Fig. 7b, d and Fig, S6a,b, page**
 **11-12, Lines 275-294**).

**Fig. R2. JUNB deficiency severely impairs hPSC differentiation into HECs**
 **and HPC.**

**a, b** JUNB KO hPSC lines were verified by Sanger sequencing (a) and western blot (b). **c**
 Representative flow cytometry density plots of HEC (CD34⁺ CD184⁺ CD73⁻) population on
 69 day 6 in WT and JUNB KO cells, respectively. **d.** Representative flow cytometry density
 plots of HPC (CD34⁺ CD43⁺) on day 8 in WT and KO cells, respectively. Error bars
 represent SD. P-values were calculated using Student's t-test, *p-value < 0.05, **p-value
 < 0.01, ***p-value < 0.001.

Following the reviewer's suggestion, we also constructed two rescue cell lines
 by inserting a DOX inducible JUNB cassette into JUNB KO hPSCs (Fig. R3a).
 To determine the roles of JUNB at different stages of hematopoietic
 specification, we induced its expression at different time windows. JUNB
 becomes highly expressed on day 3 of the HPC differentiation when the cells
 are primed to become EPCs and HECs. So, we added DOX from day 3 to day
 5 to examine the function of JUNB in HEC formation (Fig. R3a). Similarly, to
 investigate the role of JUNB in EHT, we added DOX from late day 5 to day 6
 (24h), when EHT is underway (Fig. R3a).

The results show that re-introducing JUNB during day 3-5 increased the
 percentage of HECs (about 25%) and HPCs (about 10%) compared to the
 JUNB KO cells (about 15% and 4%, respectively), which confirms the
 importance of JUNB in HEC and subsequent HPSC formation (Fig. R3b, c).
 Furthermore, adding JUNB at the EHT window (day 6) also significantly
 elevated HPC percentage (Fig. R3c), indicating HECs can effectively undergo
 EHT upon JUNB compensation. Thus, the rescue experiments verify that JUNB
 is essential for HPC formation by promoting HEC specification and EHT. **We**
 **have added these results in the revised manuscript (Fig. 7g-j, page 13,**
 **Lines 309-320).**

 **Fig. R3. Rescue of JUNB expression during hPSC differentiation into HPC.**

**a.** Schematics showing the DOX inducible JUNB expression constructs (upper panel) and
 the overview of the DOX treatment strategies during HPC differentiation (lower panel).
 DOX was added during days 3-5 or 6 to induce JUNB during HEC formation or the EHT
 process. **b.** Bar plot showing that DOX-induced ectopic JUNB expression from day 3 to
 98 day 5 rescues HEC (CD34⁺ CD73⁻ CD184⁻) percentage. **c.** Bar plot showing that induction
 of JUNB expression with DOX during either HEC specification or EHT stage increase HPC

(CD34⁺ CD43⁺) percentage significantly. P-values were calculated using Student's t-test,
*p-value < 0.05, **p-value < 0.01, ***p-value < 0.001.

**POINT-TO-POINT RESPONSES:**

**Reviewer #1**

The manuscript by Chen et al. describes the molecular characterization of
hematopoietic specification of human pluripotent stem cells (hPSCs). Despite
several studies have described the development of haematopoietic lineages
from hPSCs, the detailed mechanisms regulating the emergence of
haematopoietic cells have not been elucidated. This translates to an overall
poor efficiency of hPSC haematopoietic differentiation. Therefore, the question
addressed is timely and of interest, as a better characterization of
haematopoietic specification is needed and could, in principle, be exploited for
the generation of transplantable haematopoietic cells from hPSC as well as for
*in vitro* disease modelling. The results are interesting, however in my opinion
there are some limitations that are left unanswered and need to be addressed
to grant a publication in Nature Communication.

**Response:** We appreciated Reviewer #1 for the supportive comments!

**Specific points:**

This reviewer understands the choice of focusing on bivalent genes. However,
bivalence seems a poor predictor of a cell state when compared to ATAC-seq.
Many genes, including JUNB which is the focus of the last part of the paper, is
bivalent already in hPSCs a totally irrelevant stage for hematopoietic
specification, but it is not expressed until the endothelial stage. Given this, this
reviewer finds the analysis of bivalent genes in hPSCs that show a dramatic
activation or repression in HSPCs not extremely relevant or meaningful. Since
the authors have collected already all the data, it would be much better to
analyze at differences between the different steps, so to map the
developmental changes that occur throughout the hematopoietic specification.
In other words, what changes between the hPSC and mesoderm stage?
Between mesoderm and endothelial cells? Between endothelial cells and
HSPCs? Focusing the comparison between successive cell states will very
likely uncover more relevant regulators and be much more helpful to the
community.

**Response:** We thank the reviewers for bringing up this precious suggestion,
and we have now significantly expanded our study. Following the reviewer's
advice, we have made great efforts to analyze the differences between the
different steps, which has now led to several important discoveries.

To analyze the chromatin state across the course of differentiation, we
 performed ChromHMM analysis² based on the profiles of two histone
 modifications and identified four chromatin states: H3K4me3-only, bivalent,
 unmarked, and H3K27me3-only states, respectively (Fig. R4a, b). The
 annotation results show the H3K4me3-only regions and bivalent regions are
 enriched in the transcription start sites (TSS) and promoter regions (Fig. R4b)
 across all stages. Although both bivalent and H3K4me3-only regions have
 similar chromatin accessibilities, the expression levels of bivalent genes are
 lower, consistent with the fact that genes marked by bivalent histone
 modifications are primed to be activated³ (Fig. R4b,c). In contrast, both
 unmarked and H3K27me3-only regions have low gene expression levels and
 chromatin accessibility (Fig. R4b, c).

**Fig. R4. Dynamics of histone modification landscapes during HPC**
 **differentiation.**

**a.** Heatmap showing four chromatin states inferred from ChIP-seq datasets based on
 ChromHMM algorithm. Each row corresponds to a different histone mark, and each
 column corresponds to a different chromatin state. Regions with low H3K4me3 or H3K27me3
 modifications are labeled unmarked, whereas regions with both H3K4me3 and H3K27me3

are labeled bivalent. Darker colors indicate higher probabilities. **b.** Heatmaps showing
genomic distributions of four chromatin states in each stage during differentiation. **c, d.**
Boxplots showing gene expression levels (**c**) and chromatin accessibilities (**d**) of four
chromatin states in each stage during differentiation. FPKM, fragments per kilo-base of
transcript per million reads mapped.

Bivalent domains are often found at promoters of developmental genes⁴.
Information about the dynamic change of bivalent genes during differentiation
can aid the identification of lineage regulators^{5, 6}. Thus, we analyzed the
bivalent domain profile between successive cell types during the hematopoietic
specification process. We found that the number of bivalent domains is highest
in hPSC, reflecting more developmental regulators poised in the pluripotent
state than other progenitor cell types (Fig. R5a).

We next examined the dynamics of bivalent domains and their associated gene
expression levels during HPC differentiation. We enumerated the 4 modes of
changes in bivalent domains and calculated the ratio of each mode (Fig. R5b,
c). During hPSC differentiation into VME, about 20% of bivalent domains lose
H3K27me3 marks but gradually gain H3K4me3 levels and turn into H3K4me3-
only regions. Correspondingly, their related genes get activated (Fig. R5d). GO
analysis of these activated genes show that the most significant terms include
mesoderm development ($P < 1 \times 10^{-7}$) and response to BMP ($P < 1 \times 10^{-8}$) (Fig.
R5e), which reflect under the stimulation of BMP4 and CHIR, bivalent genes
involved in mesoderm specification (e.g., GATA6, HAND1, and BMP2) are
rapidly activated (Fig. R5f). Similarly, from VME to EPC, nearly 20% of bivalent
domains turn into H3K4me3-only regions. Prominent GO terms for these genes
related to endothelium development (Fig. R5g-i). Interestingly, genes involved
in hematopoietic progenitor cell differentiation are also induced and activated
in the EPC stage, which suggest the hematopoietic program is already primed
in the EPC as indicated by the activation of MYB (Fig. R5i). As a result, only a
few bivalent domains (about 7%) turn into H3K4me3-only states from day 5
EPC to day 8 HPC (Fig. R5c). These analyses suggest that the key bivalent
genes encoding stage-specific regulators are resolved and activated
appropriately to promote cell fate transitions during HPC differentiation.

**Fig. R5. Dynamics of active bivalent domains during hPSC to HPC**
 **differentiation.**

**a.** Bar chart showing the number of bivalent domains at each cell population. **b,c.**
 Schematic diagram of bivalent domains dynamic changes (**b**) and the corresponding
 ratios(**c**) between successive cell state transition. **d.** Boxplots showing the dynamic

changes of H3K4me3 (left), H3K27me3 (middle), and gene expression (right) of activated
bivalent genes during hPSC to VME transition. **e.** GO term analyses of active bivalent
genes during the hPSC to VME transition. **f.** The UCSC browser views show H3K4me3
and H3K27me3 modification profiles during hPSC to VME transition. The promoter regions
are shaded. The normalized RNA-seq FPKM for each gene at different stages is shown
on the left. **g.** Boxplots showing the dynamic changes of H3K4me3 (left), H3K27me3
(middle), and gene expression (right) of activated bivalent genes during the VME to EPC
transition. **h.** GO term analyses of active bivalent genes during VME to EPC transition. **i.**
The UCSC browser snapshots show H3K4me3 and H3K27me3 profiles during VME to
EPC transition. The promoter regions are shaded. The normalized RNA-seq FPKM for
each gene at different time point stages are shown on the left. The view scale of the
genome browser is adjusted according to the global data range.

Between every two successive steps, many bivalent domains remain covered
by H3K4me3 and H3K27me3 signals, and the related genes remain silenced
(Fig. R6a-c). These genes are mainly involved in non-hematopoietic lineage
commitment, such as embryonic organ development and neuron fate
commitment (Fig. R6d). Thus, the stable bivalent modifications can safeguard
the hematopoietic lineage commitment in the *in vitro* differentiation system.
However, HOXA5, HOXA9, and HOXA10, which play critical roles in definitive
HSC generation and proliferation^{7,8}, are also stable bivalent genes throughout
the *in vitro* HPC differentiation (Fig. R6e). In addition, the chromatin around
these genes remains largely inaccessible (Fig. R6f), suggesting a lack of
activating factors specific to the HOXA gene cluster during the differentiation
process. Therefore, additional efforts should be made to precisely regulate the
repression and activation of bivalent genes to further optimize the *in vitro*
hematopoietic system.

Collectively, the analysis of bivalent genes provided insights into the unique
features of transcriptional and epigenetic regulation of *in vitro* HPC formation.
**We have added these results in the revised manuscript (Fig. 3 and Fig. S3,**
**page Lines 146-190).**

**Fig. R6. Dynamics of stable bivalent domains during hPSC-HPC**
 **differentiation.**

**a-c.** Boxplots showing H3K4me3 (left), H3K27me3 (middle) signal intensities and their
 associated gene expression levels (right) of stable bivalent marked regions during the
 processes of hPSC to VME (a), VME to EPC (b), and EPC to HPC (c). The signal densities
 are calculated as H3K4me3 and H3K27me3 ChIP-seq RPKM (reads per million reads per
 231 kb). **d.** GO terms of stable bivalent domain associated genes. **e, f.** The UCSC browser
 snapshots show H3K4me3 and H3K27me3 modification profiles of the selected genes.
 The promoter regions are shaded. The normalized RNA-seq FPKM for each gene at
 different time points are shown on the left. The view scale of the genome browser is
 adjusted according to the global data range.

The authors made the effort to compare *in vitro* derived hemogenic endothelial
 cells (HECs) with those found in the embryo. However, they only cherry-picked
 a particular stage of human embryonic development, CS13. This seems to be
 unfair, as the authors are clearly aware that there are other HECs which are
 thought to be devoid of HSC potential. As such, the authors should reperform
 similarity analysis of their cells comparing them to both CS10 and CS13 HECs.

As their hPSC-derived HECs do not express HOXA genes, these cells are likely
 reflecting extra-embryonic progenitors, which are less capable to generate
 lymphoid cells and HSC.

**Response:** We thank the reviewer for raising this important question. **We**
 **have compared *in vitro* HECs to CS10 HECs and CS13 HECs in parallel.**
 **Please refer to our response to General Comment 1 (Fig. R1) for details.**

As a reference, they should also compare *in vitro*-derived arterial cells with
 those found at CS10 and CS13 as well.

**Response:** We thank the reviewers for these comments. The comparison
 among *in vitro* AECs, CS10 AECs, and CS13 AECs in parallel also shows that
 *in vitro* AEC is more similar to CS10 AECs than CS13 AECs at transcriptome
 level (Fig. R7a). Similar to *in vivo* HEC, genes related to hypoxia are
 upregulated both in CS10 AECs and CS13 AECs (Fig.R7b). And *in vitro* AECs
 express higher levels of genes related to the cell cycle transition process (Fig.
 R7b), indicating that *in vitro* AEC is also in an active proliferation state like *in*
 *vitro* HEC, likely due to the high concentration of VEGF, FGF2, and B27
 supplements in the culture medium.

**Fig. R7. Comparison of single-cell transcriptome between *in vitro* AECs**
**and *in vivo* AECs.**

**a.** Hierarchical clustering of CS10 AEC, CS13 AEC, and *in vitro* AEC based on their
transcriptome. **b.** Heat map showing gene expression patterns of CS10 AEC, CS13 AEC,
and *in vitro* AEC (left). The enriched GO terms of top differential upregulated genes in each
cell type are listed on the right.

In addition, can the authors generate HOXA⁺ HECs or HECs with lymphoid
potential so to verify that what they have described in the current manuscript
are general principles of hematopoietic specification and is not restricted to a
HOXA- developmental program?

**Response:** We thank the reviewer for raising this question. It is an intriguing
and important question if HOXA⁺ HEC or definitive HSPC formation *in vitro* also
share similar epigenetic regulation principles. We think that many global
changes in the epigenome and the TFs involved will be similar, particularly from
hPSC to VME and VME to EPC.

Insufficient expression of HOXA genes and poor lymphoid differentiation
potential is the limitation of most *in vitro* HSPC differentiation systems. Dou et
al. used the EB differentiation method to obtain HSPC. Initially, they got HSPC
with low HOXA gene expression and presumably low lymphoid differentiation
potential. When they treated their EB-derived CD34⁺ cells with all-trans retinoic
acid (ATRA), increased expression of medial HOXA genes was observed⁹. We
think the EB-derived CD34⁺ cells in their study may be similar to our
CD31⁺CD34⁺ EPCs. Their ATAC-seq study showed that ATRA treatment
helped open up chromatin at HOXA gene clusters in CD45⁺ CD34⁺ CD90⁺
hESC-HSPCs. However, they did not test the reconstitution ability of their
HOXA gene-activated HSPCs. We think it is likely that ATRA treatment will help
remove the H3K27me3 on the bivalent HOXA genes. In our ATAC-seq, we
detected 70x10³ peaks, and 40% were at the promoters in EPC and HPC (Fig
1d and Fig S1), which is about 28 x10³ peaks. In the Dou et al. study, about
1000 peaks within 500kb to TSS were induced by the RARA agonist AM580⁹.
That much less than 28 x10³ peaks at promoter regions. Therefore, we think
that the global chromatin patterns observed in our study are likely to be general
principles.

It would be ideal to profile the epigenetic roadmap in an *in vitro* differentiation
system with definite lymphoid differentiation potential. However, to generate
definitive HSC with lymphoid potential and reconstitution ability usually need
the ectopic expression of several transcription factors in HSPCs¹⁰. Thus, the
epigenome of these HSPCs may not accurately correlate with their

transcriptome. Differentiation via the embryoid body (EB) format has also been
shown to produce definite HSC with lymphoid differentiation potential ¹¹.
However, EBs contain a very heterogeneous cell population, and it is
challenging to purify the small cell populations with definitive lymphoid
differentiation potential for epigenetic profiling. Uenishi et al. reported that fine-
tune Notch signaling with DLL1-Fc and DAPT leads to the generation of
definitive hemogenic endothelium (HE) and hematopoietic progenitors (HPs)
from hPSCs. They showed that after optimizing differentiation protocol, 1 in 14
HPs have T-cell potential in limiting dilution assay (LDA)¹². Therefore, a better
marker for definitive HECs in the human system would be beneficial to purify
this rare population of cells.

Based on our integrative epigenetic analysis and information from other
published works, we think that improving the arterial feature of HEC, hypoxia
microenvironment, and treatment with PRC2 inhibitor to remove H3K27me3 on
key TFs may improve the definitive hematopoiesis potential of hPSC derived
HECs and HPCs. Moreover, reliable markers or reporters for definitive human
HECs will be valuable tools. The definitive human HECs need to be validated
in *in vitro* differentiation and *in vivo* transplantation assays which require
considerable time. These will be the goal of our following up study but beyond
the scope of our current investigation.

The fact that hematopoietic development is dependent on activation of AP-1 TF
family is already known. In Obier et al (Development 2016), the Bonifer group
have already described part of the downstream effectors of the JUN axis during
hematopoietic development, using a different strategy. **This paper should be**
**referenced and commented.** In addition, since what is downstream of JUN is
not exactly novel, **can the authors use their thorough database to identify**
**what triggers JUN activity** (EGF, TNF or other cytokines? Hypoxia?) This
would be novel and very useful for the wide community of laboratories
differentiating hPSCs in blood cells.

**Response:** We thank the reviewer for pointing this out. We have now added
the citation to our discussion (**Page 15 Lines 369-371, revised manuscript**).

Our transcriptome data reveal that JUNB is induced at the EPC stage (Fig. 6a).
To check whether JUNB is activated by VEGF or bFGF, or both. We treated
cells with only bFGF, VEGF, or both for 8 hours. The results show that VEGF
signaling, but not bFGF signaling, activates JUNB expression (Fig. R8a).
JUNB transcript level is significantly higher in single ECs from a 3D Scaffold
differentiation protocol (our unpublished study) than in single ECs from the
current monolayer differentiation protocol (Fig. R8b). As the 3D Scaffold

creates a more hypoxia environment (according to the transcriptome analysis)
 than monolayer culture, we think hypoxia may also elevate JUNB expression.

**Fig. R8. The expression of JUNB with and without VEGF treatment.**

**a.** Bar chart showing JUNB expression levels under bFGF and VEGF induction. P-values
 were calculated using Student's t-test, *p-value < 0.05, **p-value < 0.01, ***p-value < 0.001.
 p-value > 0.05 is indicated by "ns" for not significant. **b.** JUNB expression in single ECs
 from monolayer differentiation (data from this study) and in single ECs differentiated in 3D
 Scaffold (our unpublished study).

When exactly JUNB plays a role in hematopoietic specification of hPSCs? No
 hematopoietic lineages are generated from JUNB KO but is unclear whether
 this is because HECs are absent and/or unable to make the transition to blood
 cells. Can the authors perform rescue experiments, overexpressing JUNB at
 the two critical stages (HEC specification and EHT) to see when it is required?

**Response:** We thank the reviewer for raising this important question. **We have**
 **now performed rescue experiments, and our results show that both at the**
 **HEC formation stage and EHT stage, the induction of JUNB successfully**
 **rescued HPCs generation. Please refer to our response to General**
 **Comment 2 (Fig. R2) for details.**

**Minor points:**

- The authors claim that CD44 expression is regulated directly by JUNB. But
 CD44 is also highly expressed in arterial cells and the CD184⁺ fraction
 representing cells with an arterial fate are present in JUNB KO differentiating
 cells. Is CD44 expression absent in the CD184⁺ cells as well or the lack of CD44
 expression in JUNB KO cells is specific to HECs?

**Response:** We thank the reviewer for this question. The previous study had
 shown that CD44 is expressed in arterial ECs (AECs) and HECs but seldom in
 venous ECs in the early human embryo¹. JUNB is in all ECs, albeit with higher
 expression in HECs. We believe that CD44 should not be exclusively controlled
 by JUNB. We tested CD44 expression by qPCR, and the results show CD44 is

slightly down-regulated in JUNB KO AECs but not significantly (Fig. R9a). This
may indicate that JUNB is not essential for CD44 expression in AECs. JUNB
ablation leads to a 2-fold drop in CD44 gene expression compared to the WT
HECs (Fig. 7e). Our JUNB CUT&Tag result in HECs shows that JUNB binds to
the promoter of CD44 (Fig. R9b). Together, these results indicate that the
transcription of CD44 is partially controlled by JUNB in HECs. While in AEC,
where CD44 levels are lower, the activity of JUNB is not required for CD44
expression.

A previous study found that the CD44 level increased significantly during EHT¹³.
Therefore, JUNB may be required for the upregulation of CD44 in HECs
undergoing EHT.

**Fig. R9. The expression of JUNB with and without VEGF treatment.**

**a.** Bar plot showing the expression levels of CD44 in WT AEC and JUNB KO AEC. **b.** The
UCSC browser snapshots show the binding for JUNB at the genetic loci of CD44. Promoter
regions are highlighted in yellow. P-values were calculated using Student's t-test, *p-value
< 0.05, **p-value < 0.01, ***p-value < 0.001, p-value > 0.05 is indicated by "ns" for not
significant.

- Since HSCs are not generated via the protocol used in these studies, remove
"S" from HSPC and refer to those cells as HPCs.

**Response:** We thank the reviewer for this suggestion, and we have substituted
all the "HSPC" with "HPC" in the revised manuscript.

- KDR is the correct gene symbol for FLK1

**Response:** We thank the reviewer for pointing this out, and we have changed
FLK1 to KDR in the revised manuscript and figures.

- line 290: HAEC are human and not hemogenic arterial endothelial cells.

**Response:** We thank the reviewer for pointing this out. We sincerely apologize
for this mistake, and we have corrected it in the revised manuscript.

- There are several typos and language issues in the manuscript. Please
proofread carefully to correct these, taking care of homogenizing the use of past
and present tenses throughout the manuscript.

**Response:** We thank the reviewer for pointing this out. We sincerely apologize
for these mistakes, and we have corrected them across the revised manuscript.

**Reviewer #2 (Remarks to the Author):**

The paper by Chen et al. describes epigenomic and transcriptomic analysis of
cell populations emerging during hematopoietic differentiation of H1 hESCs. By
analyzing hESC bivalent genes which get active during hematopoietic
differentiation, authors discovered that JUNB has a bivalent promoter in hESCs
and get activated in endothelial and hematopoietic cells. To find out whether
JUNB has effect on hematopoietic differentiation, JUNB knockout hESCs were
generated. These knockout cells failed to produce blood. By identifying JUNB
as a master regulator of hematopoietic commitment in hESC differentiation
culture this paper makes a novel contribution to our understanding of
transcriptional program regulating hematopoietic development.

**Response:** We thank the reviewer for the encouraging comments and
appreciate the values and significance of our work.

**Comments:**

1. To increase confidence in the obtained JUNB results and eliminate a
possibility of off-target effects, authors should demonstrate if similar results can
be obtained using several JUNB knockout clones. In addition, rescue
experiments should be performed to show a restoration of hematopoietic
potential in JUNB knockout cells following introducing exogenous JUNB.

**Response:** We thank the reviewer for raising this important question. **We have**
**now repeated the experiments with another JUNB knockout clone and get**
**the consistent results. Follow the reviewer's suggestion, we also have**
**performed rescue experiment and our results show that both at the HEC**
**formation stage and EHT stage, the induction of JUNB successfully**
**rescued HPCs generation. Please refer to our response to General**
**Comment 2 (Fig. R2) for details.**

2. What type of hematopoiesis produced in this system, extraembryonic or
intraembryonic? Does JUNB affect intraembryonic or extraembryonic-type
hematopoiesis or both? What types of CFU this protocol produces? Do CD34+
cells generated in this protocol possess lymphoid potential?

**Response:** We thank the reviewer for raising these critical questions. We have
performed PCA analysis of the transcriptome of our HPC and published
datasets. The HPCs (CD34⁺CD43⁺) produced in our study is the most similar
to HSPC differentiated using the EB method (Fig. R10a). They also have a
closer resemblance to the fetal liver (FL) HSC and umbilical cord blood (USC)
HSC than to the aorta-gonadal-mesonephros (AGM) HSC. Therefore, they are
skewed towards extraembryonic hematopoiesis (Fig. R10a). In our system, we
found that JUNB plays essential roles in HEC formation and the EHT process,
which are shared by extraembryonic and intraembryonic-type hematopoiesis.
In addition, the results from JUNB knockout (KO), overexpression (OE), and
CUT&Tag experiments demonstrated that it regulates many key intraembryonic
hematopoiesis genes such as RUNX1 and GATA2 (Fig. R10b). Therefore, we
believe JUNB affects both intraembryonic and extraembryonic-type
hematopoiesis.

Following reviewer 2's suggestion, we did the colony-formation unit (CFU)
experiments with our HPCs (CD43⁺CD34⁺). The results show that our HPCs
can form typical erythroid (CFU-E and BFU-E), granulocyte (CFU-G),
macrophage (CFU-M), granulocyte–macrophage (CFU-GM), and multi-lineage
(CFU-GEMM) colonies, which are primarily myeloid lineage cell types (Fig.
R10c). We have temped to induce CD34⁺ HPCs to differentiate into T cells but
did not detect CD4⁺ and CD8⁺ T cells. Thus, our HPCs seem to have poor
lymphoid potential.

**Fig. R10. Characterization of hematopoiesis produced in this system.**

**a.** Principal component analysis (PCA) of HPC and HSPC samples based on their
 transcriptome. **b.** The UCSC browser snapshots show the binding for JUNB at the genetic
 loci of RUNX1 and GATA2. Promoters are highlighted in yellow. **c.** The pictures show the
 results of the colony-forming assay for day 8 HPCs. Scale bars, 100 μ m. AGM: aorta-
 gonad-mesonephros, CFU: colony-forming unit, BFU: burst-forming units, E: erythroid, EB:
 embryoid body, FL: fetal liver, UCB: umbilical cord blood, M: macrophage, G: granulocyte,
 GM: granulocyte-macrophage, GEMM: Granulocyte-Erythrocyte-Monocyte/macrophage-
 Megakaryocyte.

**3. Please describe experimental design for experiments depicted in Fig.4. What**
 **was the starting population for these experiments, isolated CD34⁺ cells?**

**Response:** We thank the reviewer for pointing this out. The experimental
 design is illustrated in Fig. R11.

In our protocol, hPSCs were first treated with BMP4 and CHIR99021 for 3 days
 to induce KDR⁺ vascular mesoderm cells (VMEs). The cells were then re-plated
 in a medium supplemented with VEGF and bFGF for 2 days to induce CD34⁺

endothelial progenitor cells (EPCs). Afterwards, SB431542 was added to
promote EHT to generate CD43⁺ CD34⁺ HPCs.

We took all the differentiating cells treated with SB431542 for 6 h on day 6 for
single-cell RNA-seq analysis. This cell mixture contains EPCs, HEC
undergoing EHT, and newly formed HPC, together with other mesoderm cell
types. Analyzing this cell mixture by scRNA-seq will reveal different cell types
in the culture during the EHT window and provide information about potential
cell-cell interactions. **We have added this diagram to Fig. 4a of the revised**
**manuscript.**

**Fig. R11. Schematic representation of the sampling collection for the**
**scRNA-seq.**

Schematic showing the sequential processes of HPCs differentiation from hPSCs through
the specification of mesoderm cells (VME, KDR⁺), formation of the CD34⁺ endothelial
progenitor cells (EPCs), and HPCs generation from CD34⁺ HEC via EHT process.

4. How hemogenic endothelial clusters and HPC clusters were identified? What
are the differences in HPC-T1 and T2 clusters? Please provide in supplement
RNAseq UMAP plots with marked RUNX1, CD44, SOX17, CDH5, and CD34
expression.

**Response:** We thank the reviewers for raising these critical questions.

Hemogenic endothelial cells (HECs) are specialized endothelial cells
expressing endothelial and hematopoietic genes and are a relatively rare
population both *in vivo*^{1, 14} and *in vitro*¹⁵.

To identify HEC clusters, we first identified epithelial (Epi), mesenchymal (Mes),
endothelial (EC), and hematopoietic progenitor cell (HPC) clusters from the
whole cell population based on the transcriptional signature of each cell type
(Fig. R12a). Next, the annotated EC and HPC populations were extracted, re-
normalized, and separated into 5 sub-clusters (Fig. R12b). Among them, the
sub-cluster expressing both endothelial genes (such as *GJA4*, *CD44*) and
hematopoietic genes (such as *CLEC11A*) were annotated as the hemogenic
endothelium (HE) cluster. The sub-cluster expressing hematopoietic genes
such as *CLEC11A*¹⁶ was annotated as pre-HPC clusters. Interestingly, we
found the pre-HPC cluster could be divided into two groups based on the

expression level of cell cycle genes. Pre-HPC Type II (pre-HPC TII) group has
 more cycling cells in either S or G2/M phase than the pre-HPC Type I (pre-HPC
 TI) group. Besides, trajectory analysis revealed that the pre-HPC TII cluster has
 a closer relationship to HE than the pre-HPC TI. These results suggest that pre-
 HPC TII cells may be newly emerged hematopoietic cells which have just
 completed the EHT process. Therefore, we use pre-HPC to refer to these early
 HPCs at this stage (**Fig. 4, Fig.6, Lines 218-223, revised manuscript**).

Fig. R12.b showed the expression levels and distributions of RUNX1, CD44,
 SOX17, CDH5, and CD34. SOX17, CDH5, and CD34 are more expressed by
 AEC TI AEC TII cells on the left side, while RUNX1 and CD44 were higher in
 pre-HPC TI and pre-HPC TII, and HE cells. **This result is added to the revised
 manuscript Fig. S4c)**

**Fig. R12. Single-cell transcriptomic analysis of differentiating cells at**
 **day6.**

**a.** UMAP clustering plots showing that all cells are grouped into 4 clusters: Epi,
 and HPC (left panel). The annotated EC and HPC were extracted, re-analyzed,
 and separated into 5 sub-clusters: AEC TI, AEC TII, pre-HPC TI, pre-HPC TII,
 and HE (right panel). **b.** The scatter plots show the expression and distribution of
 SOX17, CDH5, RUNX1, CD44 and CD34. Epi: Epithelial cell, Mes: Mesoderm cell,
 EC: Endothelial Cell, HPC: Hematopoietic Progenitor Cell. AEC TI: AEC Type I,
 AEC TII: AEC Type II, HPC TI: HPC Type I, HPC TII: HPC Type II, HE, hemogenic endothelium.

**5. Authors found that hemogenic endothelium generated in hPSC cultures is**
 **highly proliferative. What about CS13 HECs?**

**Response:** We thank the reviewer for this question. We performed cell cycle
 analysis using the scRNA-seq data of CS13 HEC and found they are not as
 proliferative as *in vitro* HEC. Please refer to our response to General Comment
 1 (Fig. R1) for details.

6. In introduction, authors describe just two waves of embryonic hematopoiesis
and failed to acknowledge its complexity and multiple waves (see DOI:
10.1038/nrm.2016.127).

**Response:** We thank the reviewer for pointing this out. We sincerely apologize
for this mistake and have revised the text in the introduction part as shown
below **in our revised manuscript (page 2, Lines 33-42).**

Blood development in mammalian embryogenesis involves three waves of
spatiotemporally distinct hematopoiesis^{17, 18}. The first and second waves arise
in the yolk sac and are considered extra-embryonic hematopoiesis¹⁸. The first
wave is transitory and mainly produces primitive erythrocytes, supporting tissue
oxygenation for the growing embryo^{18, 19}. The second wave gives rise to
multipotent progenitors, with erythro-myeloid progenitors (EMPs) and
lymphoid-primed progenitors (LMPP), independent of hematopoietic stem cells
(HSCs)^{20, 21}. The third wave is intra-embryonic hematopoiesis, where definitive
HSCs emerge from the dorsal aorta of the aorta-gonad-mesonephros (AGM)
region²², and are capable of engrafting adult recipients²³. In all three waves,
HSCs are developed from a group of specialized hemogenic endothelial cells
(HECs) via the endothelial-to-hematopoietic transition (EHT)¹.

7. Authors claim that JUNB knockout did not impair the generation of CD34⁺
EPCs. However, CD34 is broadly express in non-endothelial cell types. To
ensure that this statement is correct, additional endothelial markers, such as
VE-cadherin and CD31 should be evaluated in WT and KO cultures.

**Response:** We thank the reviewer for pointing this out. We checked endothelial
markers CD31 and CD144, the percentage of CD31⁺ and CD144⁺ cells are
similar in WT (45.7%, 36.5%), JUNB KO1 (47.3%, 36.2%) and JUNB KO2
(44.4%, 33.7%) (Fig. R13). These results support our conclusion that the
generation of CD34⁺ EPCs is not affected by JUNB KO. **We have added these**
**data in the revised manuscript (Fig. S6f, page 12, Lines 282-284).**

**Fig. R13. Characterization of CD34⁺ EPC.**

**a, b.** Density plots showing flow cytometry results of CD34⁺ endothelial progenitor cells
 (EPC) for the surface markers CD31(**a**) and CD144(**b**). Positive cell percentages are
 labeled. And quantification results are shown on the right. P-values were calculated using
 Student's t-test, *p-value < 0.05, **p-value < 0.01, ***p-value < 0.001, p-value > 0.05 is
 indicated by "ns" for not significant.

**Minor:**

1. Ref 5 and 6 are related to EHT in AGM region and are not related to EHT
 during primitive hematopoiesis.

**Response:** We thank reviewer 2 for careful reading and have replaced Ref 5
 and 6 with new Ref 8 in the revised manuscript.

2. Ref. 7 is incorrect. This reference describes the effect of VEGF and FGF2
 on HUVECs and has nothing to do with mesodermal differentiation.

**Response:** We thank the reviewer for pointing this out, and we have now
 removed Ref 7 and added new Ref 9 in the revised manuscript.

3. Line 50: hematopoietic endothelium should be hemogenic endothelium.

**Response:** We sincerely apologize for the mistake and have corrected this typo
 in the revised manuscript.

4. In result section, please introduce hPSC line used in this study (H1 hESC).

**Response:** We thank the reviewer for pointing this out. We have added a brief
 introduction of H1 hESC, as shown below. **We have also added this part to**

**the revised manuscript (Lines 75-77 and Lines 406-412, revised**
**manuscript).**

The H1 cell line was obtained from WiCell and routinely maintained on MEF
feeders in the hESC medium: KnockOut DMEM culture medium supplemented
with 20% (vol/vol) KnockOut serum replacement, 1% nonessential amino acids
(NEAA), 1 mM L-GlutaMAX-I, 0.1 mM β -mercaptoethanol, and 8 ng/mL bFGF.
They were passaged with 1 mg/mL collagenase IV (Invitrogen) and seeded
onto a 25 cm² flask that had been pre-coated with 0.1% gelatin solution (Sigma
Aldrich). For differentiation, hESCs were maintained on vitronectin or Matrigel
(BD Biosciences)-coated plates (Corning) in E8 medium (STEMCELL
Technologies). We have now added these descriptions to the revised
manuscript.

5. Line 140. SOX17 is involved in EHT, the major function of this gene is to
promote arterial commitment.

**Response:** We thank the reviewer for the correction, and we have revised this
in the revised manuscript (**page 6, Lines 137-138**).

6. Line 168. Correct H3K37me3 typo.

**Response:** We sincerely apologize for this mistake. We have now corrected
this typo.

7. Line 273. FLK1 differences are negligible and not significant. Word
"noticeable" should not be used. Please use the current KDR nomenclature for
FLK1.

**Response:** We thank the reviewer for pointing this out, and we have changed
FLK1 to KDR in the revised manuscript.

**Reviewer #3 (Remarks to the Author):**

To understand the mechanism of HSPC fate determination in humans, the
authors dissect the epigenomic roadmap from hPSCs to HSPCs by profiling
chromatin accessibility, histone modifications and transcriptome. Generally, the
epigenetic feature dynamics and gene expression dynamics are highly
correlated during differentiation. For the chromatin accessibility, the regulatory
regions become accessible before key TF binding to the chromatin. For the
histone modifications, the bivalent genes are characterized by stage-specific
H3K4me3 and H3K27me3 during HSPC differentiation. Specifically, they reveal
that EHT contains several intermediate subpopulations with unique
transcriptome and chromatin states. Furthermore, they identify JUNB as a new

regulator of HSPC differentiation and the deficiency of JUNB by iCRISPR will
impair HEC formation and EHT.

**Response:** We thank the reviewer for these comments.

**Major comments:**

1. Whether the differentiation protocol used in this study can generate
functional HSPCs with complete self-renewal and engraftment abilities remains
unknown.

**Response:** We thank the reviewer for this important comment. We have
performed transplantation of our hPSC-derived HSPCs into irradiated NSG
(NOD.Cg-Prkdc Il2rg/SzJ) mice and did not detect any reconstitution. Therefore,
they are more likely to be primitive CD43⁺ HSPCs, with low self-renewal and
reconstitution ability *in vivo*.

Although significant progress has been made in the field of hematopoietic
differentiation from hPSCs, including the establishment of multiple
hematopoietic differentiation protocols and the generation of functional blood
cells^{24, 25}, hPSC-derived HSPCs cannot reconstitute hematopoiesis in NOG-
SCID mice. The generation of definitive HSCs from PSCs has been a long-
sought goal. A study by Sugimura showed that after transduced with seven
transcription factors (ERG, HOXA5, HOXA9, HOXA10, LCOR, RUNX1, and
SPI1), hESC-derived HE cells acquire the definitive hematopoietic potential and
reconstitute hematopoiesis in NOG-SCID mice¹⁰. However, for the clinical
application of hPSC-derived HSCs, it is better to avoid the ectopic expression
of hematopoietic TFs.

As the epigenetic landscape defines the cell fate and potentials, we think a
better understanding of the epigenomic roadmap of the HSPC *in vitro*
differentiation process will bring insights into how to improve the culture
condition and help to discover new regulators and principles.

2. Related to comment #1, if the generation of HSPC with complete engraftment
ability is difficult to achieve, whether the profiling of epigenetic features and
transcriptome features in this study can resolve the bottleneck of induction of
real HSPC *in vitro*.

**Response:** We thank the reviewer for raising this question. Although we did
not solve the current bottleneck in the present study, as mentioned above, our
findings have provided new insights about the critical difference between *in vitro*
and *in vivo* HECs and HSPCs, and the directions to optimize the differentiation
protocol.

A recent scRNA-seq study profiling early embryonic hematopoiesis in human
embryos showed that the HEC group could be divided into two temporally and

molecularly distinct populations. The earlier emerging population lacks arterial
 features, while the later emerging HSC-primed HEC population shows
 distinctive arterial endothelial features²⁶. By comparing the single-cell
 transcriptomes of *in vitro* generated HEC with those generated *in vivo*, we
 identified molecular pathways and regulators as potential targets for improving
 HSPC differentiation *in vitro*. For example, genes responsible for arterial
 endothelium development (such as DLL4, SOX17) are expressed at lower
 levels in the *in vitro* produced HECs than CS13 HECs, which are considered
 HSC primed HECs (Fig. R14a). They are both bivalent genes, and the
 H3K27me3 marks are not removed in EPC and HPC stages, which may
 account for their low expression in *in vitro* HECs and HPCs (Fig. R14b).

Through inferring TF binding sites from the open chromatin landscape and
 analyzing the dynamic change of the bivalent chromatin, we uncovered
 interesting principles about cell fate transitions during HPC specification. HOXA
 genes involved in definitive HSC formation are also bivalent genes covered with
 heavy H3K27me3 marks in EPC and HPC stages (Fig. 3j), and they remain
 poised due to the repressive chromatin states. As H3K27me3 modification is
 catalyzed by the PRC2 complex, treating cells with PRC2 inhibitors may help
 to activate more definitive HSC TFs. Comparison analysis of HECs from *in vivo*
 and *in vitro* indicated that promoting arterial endothelial features, culture in
 hypoxia condition, and cell cycle adjustment may also enhance the potential of
 *in vitro* differentiated HECs to form definitive hematopoietic stem cells (HSCs).
 The chromatin analysis and transcriptome profiling also lead us to discover
 JUNB as a new regulator for HEC and HPC formation from hPSCs.

**Fig. R14. Gene expression profile and their chromatin state.**

**a.** Bar plot showing the expression levels of SOX17 and DLL4 in CS10 HEC, CS13HEC,
 and *in vitro* HEC, respectively. **b.** The UCSC browser views show the H3K4me3,

H3K27me3, and chromatin state at the SOX17 and DLL4 gene loci. The promoter regions
are highlighted in yellow.

3. Sc-RNAseq data showed that Junb is expressed in EC and HPC populations.
Functional analysis of Junb showed that it could regulate hematopoietic
specification and ChIP-seq data showed hematopoietic genes were direct
targets of JUNB. However, how JUNB regulates hematopoietic TFs specifically
remains unclear.

**Response:** We thank the reviewer for these questions. In our study, we found
that the JUNB motif is significantly enriched in the open chromatin of EPCs,
implying it may have an important function there. The RNA-seq results also
show that many HEC formation and EHT related genes are down-regulated
upon JUNB KO (Fig. 7e). We performed JUNB CUT&Tag²⁷ and the results
reveal that it can bind to the promoters of known key hematopoietic regulators
(such as RUNX1, CD44) (Fig. 7f). Thus, JUNB can directly regulate the
expression of HEC and EHT related TFs.

Several studies reported that AP-1 could function as a pioneer factor to remodel
the chromatin landscape, therefore affecting chromatin accessibility.
Subsequently, lineage-specific TFs are recruited by AP-1 to the target genes to
establish cell identities^{28, 29}. We hypothesized that JUNB might also be a
pioneer factor during HPC differentiation, making specific chromatin regions
more accessible for hematopoietic TFs.

To test this hypothesis, we performed ATAC-seq on WT and JUNB KO HECs.
We found a significant portion (20705/60761, 34.1%) of open regions are
attenuated due to ablation of JUNB (Fig. R15a). We call these regions JUNB
dependent sites. Motif analysis reveals significant enrichment of the JUNB
motif at the JUNB dependent sites (Fig. R15b). Besides, hematopoietic TF
motifs, such as ERG, GATA2, and RUNX1, are also enriched at JUNB
dependent sites (Fig. R15b). These results support the hypothesis that JUNB
may be a pioneer factor for other hematopoietic TFs.

The above results suggest that JUNB may regulate hematopoietic TFs by two
mechanisms: open up specific chromatin regions to facilitate the deposition of
hematopoietic master TFs and directly bind to the promoter of important
hematopoietic genes. **We have added these results to the revised
manuscript (Fig. 7f, Fig. S7b, c, page 12, Lines 295-308; page 15, Lines
376-379).**

**Fig. R15. The chromatin accessibility in WT- and JUNB null HEC.**

**a.** left, Venn plot showing ATAC-seq peaks in WT and JUNB KO HECs, respectively. right,
 Metaplot showing the levels of ATAC-seq signals at JUNB dependent sites in WT and
 JUNB KO HECs. Right, Metaplot showing the levels of ATAC-seq signals at JUNB
 dependent sites in WT and JUNB null HECs. **b.** Bar plot showing TF motifs enriched from
 JUNB dependent peaks. P-values are estimated by HOMER.

**4. JUNB deficiency impaired HEC and HSPC differentiation *in vitro*. Whether**
 **it can play the similar *in vivo*? Can overexpression of JUNB facilitate**
 **the generation of functional human HSCs *in vitro*?**

**Response:** We thank the reviewer for these important questions. For the first
 question, *junb* KO mice die between E7.5-10.5 and the prominent phenotype is
 poor development of yolk sac and placenta vasculature³⁰. We think it is likely
 that JUNB may regulate hematopoiesis in the early human embryo. We
 analyzed the scRNA-seq data of CS10 and CS13 HECs. JUNB is expressed at
 higher levels in *in vivo* HECs than *in vitro* HECs (Fig. R16a). Data mining from
 another scRNA-seq study revealed that JUNB is also highly expressed in *in*
 *in vivo* HSPC compared to HSPCs generated from hPSC³¹ (Fig. R16b). In the *in*
 *vitro* differentiation system, JUNB KO severely affected HEC and HPC
 formation. At the chromatin level, JUNB bind to the promoter of key
 hematopoietic TFs such as RUNX1, and JUNB KO lead to reduced open
 chromatin regions bind by many hematopoietic TFs (Fig. R15). The above
 results illustrate that JUNB is expressed at the right place and time during early
 human embryo hematopoiesis. The loss-of-function study revealed that JUNB
 could impact the expression and chromatin binding of many hematopoietic
 regulators during HEC formation and the EHT window. Therefore, we speculate
 it is also involved in *in vivo* HEC and HSC formation. **We have now added this**
 **part to the discussion in the revised manuscript (Fig. 6e, Lines 388-395).**

**Fig. R16. Comparison of JUNB expression in *in vitro* and *in vivo* HEC and**
 **HSPC.**

**a.** Bar plot showing the JUNB expression levels in CS10 HEC, CS13 HEC, and *in vitro*
 HEC. **b.** Heatmap showing JUNB expression levels in *in vitro* HSPC and *in vivo* HSPC

(Modified from Figure 5E in Fidanza et al. a)³¹.

For the second question, we constructed doxycycline (DOX) inducible JUNB
 overexpression H1 hESC line. DOX was added from differentiation day 3 to
 induce JUNB expression (Fig. R17a). The percentage of HEC and HPC
 increases only slightly compared to WT cells. We also transplanted JUNB OE
 HPCs into 10 irradiated NSG (NOD.Cg-Prkdc Il2rg/SzJ) mice but did not detect
 any reconstitution. We speculate that JUNB may also be a context-dependent
 pioneer factor in making nearby recognition sites accessible for master
 hematopoietic TFs. It might be worthwhile to co-express JUNB with other
 hematopoietic TFs to see whether this can improve the efficiency of HPC
 differentiation *in vitro*. **We have now added these data in Fig. S7e-g, and in**
 **the discussion, page 15, Lines 381-382.**

**Fig. R17. JUNB overexpression inhibits HPC differentiation.**

**a.** Schematic illustrating the JUNB inducible expression constructs and the induction time
 window. **b, c.** Bar plots showing the percentage of HEC (CD34⁺ CD73⁻ CD184⁻) and HPC
 (CD34⁺ CD43⁺) on differentiation day 6 (**b**) and day 8 (**c**) in WT and JUNB OE cells. *p-
 value < 0.05, **p-value < 0.01, ***p-value < 0.001, p-value > 0.05 is indicated by "ns" for
 not significant.

**Minor comments:**

1. The y-axis of Fig 4h is not labeled.

**Response:** We thank the reviewer for pointing this out, and we have amended
this in the revised manuscript (**Fig. 4i, revised**).

2. The result of Fig 4g shows the developmental path of EHT, have you tried
other analysis methods, such as **RNA velocity** to validate this result?

**Response:** We thank the reviewers for this valuable suggestion. We have now
repeated this analysis using the RNA velocity algorithm³². The new result also
shows that HE is in the interface of the AEC cluster and HPC cluster, which is
consistent with the pseudotime analysis result (Fig 4h) using Monocle 2³³.

**Fig. R18. RNA velocity analysis of ECs and HPCs.**

RNA velocity fields visualized on UMAP projection of sub-clusters.

3. Since a lot of sequencing omics data have been obtained, why not building
a website to display all the omics data in a visual way so that readers can better
use this information?

**Response:** We thank the reviewer for this valuable suggestion. We have set
up to view all our datasets using the UCSC Genome Browser. The link of RNA-
seq, ChIP-seq, and ATAC-seq datasets of hPSC to HPCs is
<ftp://166.111.152.245:2100/chenxia/hubDirectory/hub.txt>. To view these tracks,
one can click into the navigation bar "My Data" and then "Track Hubs" to reach
the Track Hubs page on the UCSC Genome Browser. Then one can paste in
"ftp://166.111.152.245:2100/chenxia/hubDirectory/hub.txt" and click the
"Connect" button to see the data from our Hub. The process is as follows:

First step: click into "Track Hubs".

Second step: input
"ftp://166.111.152.245:2100/chenxia/hubDirectory/hub.txt".

Third step: click "GO".

And the link of the single-cell database is <http://166.111.152.245:3850/>, where
one can input genes of interest and see their expression in different cell clusters.

Reference:

- 1. Y Zeng, *et al.* Tracing the first hematopoietic stem cell generation in human embryo
by single-cell RNA sequencing. *Cell research* **29**, 881-894 (2019).
- 2. J Ernst&M Kellis. Chromatin-state discovery and genome annotation with
ChromHMM. *Nature Protocols* **12**, 2478-2492 (2017).
- 3. NS Christophersen&K Helin. Epigenetic control of embryonic stem cell fate. *J Exp*
*Med* **207**, 2287-2295 (2010).
- 4. BE Bernstein, *et al.* A bivalent chromatin structure marks key developmental genes in
embryonic stem cells. *Cell* **125**, 315-26 (2006).
- 5. NJ Palpant, *et al.* Chromatin and Transcriptional Analysis of Mesoderm Progenitor
Cells Identifies HOPX as a Regulator of Primitive Hematopoiesis. *Cell Rep* **20**, 1597-
1608 (2017).
- 6. SL Paige, *et al.* A Temporal Chromatin Signature in Human Embryonic Stem Cells
Identifies Regulators of Cardiac Development. *Cell* **151**, 221-232 (2012).
- 7. H Yu, *et al.* Downregulation of Prdm16 mRNA is a specific antileukemic mechanism
during HOXB4-mediated HSC expansion in vivo. *Blood* **124**, 1737-1747 (2014).
- 8. HJ Lawrence, *et al.* Loss of expression of the Hoxa-9 homeobox gene impairs the
proliferation and repopulating ability of hematopoietic stem cells. *Blood* **106**, 3988-
3994 (2005).
- 9. DR Dou, *et al.* Medial HOXA genes demarcate haematopoietic stem cell fate during
human development. *Nat Cell Biol* **18**, 595-606 (2016).
- 10. R Sugimura, *et al.* Haematopoietic stem and progenitor cells from human pluripotent
stem cells. *Nature* **545**, 432-+ (2017).
- 11. ES Ng, *et al.* Differentiation of human embryonic stem cells to HOXA(+) hemogenic
vasculature that resembles the aorta-gonad-mesonephros. *Nat Biotechnol* **34**, 1168-
1179 (2016).
- 12. GI Uenishi, *et al.* NOTCH signaling specifies arterial-type definitive hemogenic
endothelium from human pluripotent stem cells. *Nat Commun* **9**, 1828 (2018).
- 13. M Oatley, *et al.* Single-cell transcriptomics identifies CD44 as a marker and regulator
of endothelial to haematopoietic transition. *Nature Communications* **11**, 586 (2020).
- 14. LC Goldie, JL Lucitti, ME Dickinson&KK Hirschi. Cell signaling directing the formation
and function of hemogenic endothelium during murine embryogenesis. *Blood* **112**,
3194-204 (2008).
- 15. KD Choi, *et al.* Identification of the hemogenic endothelial progenitor and its direct
precursor in human pluripotent stem cell differentiation cultures. *Cell Rep* **2**, 553-67
(2012).
- 16. A Hiraoka, *et al.* Cloning, expression, and characterization of a cDNA encoding a
novel human growth factor for primitive hematopoietic progenitor cells. *P Natl Acad*
*Sci USA* **94**, 7577-7582 (1997).
- 17. A Ditadi, CM Sturgeon&G Keller. A view of human haematopoietic development from
the Petri dish. *Nat Rev Mol Cell Bio* **18**, 56-67 (2017).

- 18. A Ivanovs, *et al.* Human haematopoietic stem cell development: from the embryo to
the dish. *Development* **144**, 2323-2337 (2017).
- 19. SH Orkin&LI Zon. Hematopoiesis: An evolving paradigm for stem cell biology. *Cell*
**132**, 631-644 (2008).
- 20. C Boiers, *et al.* Lymphomyeloid Contribution of an Immune-Restricted Progenitor
Emerging Prior to Definitive Hematopoietic Stem Cells. *Cell Stem Cell* **13**, 535-548
(2013).
- 21. KE McGrath, *et al.* Distinct Sources of Hematopoietic Progenitors Emerge before
HSCs and Provide Functional Blood Cells in the Mammalian Embryo. *Cell Rep* **11**,
1892-1904 (2015).
- 22. A Medvinsky&E Dzierzak. Definitive hematopoiesis is autonomously initiated by the
AGM region. *Cell* **86**, 897-906 (1996).
- 23. AM Muller, A Medvinsky, J Strouboulis, F Grosveld&E Dzierzak. Development of
Hematopoietic Stem-Cell Activity in the Mouse Embryo. *Immunity* **1**, 291-301 (1994).
- 24. II Slukvin. Generating human hematopoietic stem cells in vitro - exploring endothelial
to hematopoietic transition as a portal for stemness acquisition. *Febs Lett* **590**, 4126-
4143 (2016).
- 25. R Sugimura, *et al.* Haematopoietic stem and progenitor cells from human pluripotent
stem cells. *Nature* **545**, 432-438 (2017).
- 26. Y Zeng, *et al.* Tracing the first hematopoietic stem cell generation in human embryo
by single-cell RNA sequencing. *Cell Res* **29**, 881-894 (2019).
- 27. HS Kaya-Okur, *et al.* CUT&Tag for efficient epigenomic profiling of small samples
and single cells. *Nature Communications* **10**, (2019).
- 28. RI Martinez-Zamudio, *et al.* AP-1 imprints a reversible transcriptional programme of
senescent cells. *Nat Cell Biol* **22**, 842-855 (2020).
- 29. RI Sherwood, *et al.* Discovery of directional and nondirectional pioneer transcription
factors by modeling DNase profile magnitude and shape. *Nat Biotechnol* **32**, 171-178
(2014).
- 30. M Schorpp-Kistner, ZQ Wang, P Angel&EF Wagner. JunB is essential for mammalian
placentation. *Embo J* **18**, 934-948 (1999).
- 31. A Fidenza, *et al.* Single-cell analyses and machine learning define hematopoietic
progenitor and HSC-like cells derived from human PSCs. *Blood* **136**, 2893-2904
(2020).
- 32. G La Manno, *et al.* RNA velocity of single cells. *Nature* **560**, 494-+ (2018).
- 33. C Trapnell, *et al.* The dynamics and regulators of cell fate decisions are revealed by
pseudotemporal ordering of single cells. *Nat Biotechnol* **32**, 381-U251 (2014).

REVIEWERS' COMMENTS

Reviewer #1 (Remarks to the Author):

The authors have done extensive revisions, including new analysis and experiments as well as clarification of the data presentation and message. They have addressed the primary concerns of this reviewer. I am satisfied with the revised paper which is clearly improved.

Reviewer #2 (Remarks to the Author):

All my concerns are adequately addressed. I have minor comment regarding defining the hemogenic endothelium as CD184- cells. Indeed, the emerging hemogenic endothelial cells do not express CXCR4 or other arterial markers. However, CXCR4 and DLL4 arterial markers became expressed in hemogenic endothelium following acquisition arterial features. As shown in mice and human PSC system arterialized CXCR4+ hemogenic endothelium possess strong HSC and lymphomyeloid potentials <https://pubmed.ncbi.nlm.nih.gov/34525376/> , <https://pubmed.ncbi.nlm.nih.gov/35332125/>, <https://pubmed.ncbi.nlm.nih.gov/29791856/>, <https://pubmed.ncbi.nlm.nih.gov/33596423/>

Reviewer #3 (Remarks to the Author):

The authors have adequately addressed all of my previous concerns, I congratulate them for the nice work, and the revised manuscript now is suitable for publication in Nature Communication.